# Convection enhanced delivery of Rhenium (¹⁸⁶Re) Obisbemeda (¹⁸⁶RNL) in recurrent glioma: a multicenter, single arm, phase 1 clinical trial

Andrew J. Brenner [1] ✉, Toral Patel [2], Ande Bao [3], William T. Phillips [1], Joel E. Michalek[1], Michael Youssef[2], Jeffrey S. Weinberg [4], Carlos Kamiya Matsuoka [4], Marc H. Hedrick[5], Norman LaFrance[5], Melissa Moore [5] & John R. Floyd[1]

Rhenium (¹⁸⁶Re) Obisbemeda (¹⁸⁶RNL), chelated-¹⁸⁶Re encapsulated in nanoliposomes and delivered to brain tumors via convection enhanced delivery (CED), was evaluated in a Phase 1 dose escalation trial (NCT01906385). The primary objective was to determine the maximum tolerated dose (MTD). Secondary objectives included safety and tolerability, dose distribution, the overall response rate (ORR), disease-specific progression-free survival (PFS), and overall survival (OS). 21 patients received up to 22.3 mCi ¹⁸⁶RNL over 6 dosing cohorts. Most adverse events (AEs) were unrelated to ¹⁸⁶RNL and the MTD was not reached. Although not predefined outcomes, the mOS and mPFS were 11 and 4 months, respectively, and found to correlate with radiation absorbed dose to the tumor and percent tumor treated. When dichotomized by absorbed dose of 100 Gy, the mOS and mPFS were 17 months and 6 months, respectively, for >100 Gy, compared to 6 (mOS) and 2 (mPFS) months, respectively, for <100 Gy. For ORR, 57.1% exhibited stable disease (SD), 4.8% partial response, and 38.1% progressive disease. Overall, patients received radiation absorbed doses without significant toxicity higher than possible with external beam radiation therapy (EBRT) and demonstrated mOS beyond standard of care for recurrent glioblastoma (~8 months).

Glioblastoma is the most common primary brain tumor diagnosed in adults[1,2], with >90% having recurrence at the original location[3]. Front-line treatment includes maximal surgical resection, adjuvant radiation therapy with concurrent temozolomide, and 6 months of single agent temozolomide with tumor treatment fields[4,5]. Standard of care (SoC) for patients with recurrent glioblastoma is not well defined[6]. Median overall survival (mOS) with first progression is typically ~7.4–9.2 months[7,8] and with second progression, ~4 months[9]. As a disease with a pattern of recurrence, resistance to chemotherapies, and difficulty to treat, durable treatments that can directly target the tumor while sparing healthy tissue remain an unmet need.

While external beam radiation therapy (EBRT) is an essential component of tumor treatment, its dosage in glioblastoma is restricted to ~60 Gy over multiple fractions to limit toxicity to surrounding

¹Mays Cancer Center at UT Health San Antonio, San Antonio, TX, USA. ²UT Southwestern Medical Center of Dallas, Dallas, TX, USA. ³Case Western Reserve University, Cleveland, OH, USA. ⁴University of Texas MD Anderson Cancer Center, Houston, TX, USA. ⁵Plus Therapeutics, Austin, TX, USA. ✉e-mail: BrennerA@uthscsa.edu

normal brain tissue[10]. Recently, therapeutics with beta- and alpha-emitting radionuclides attached to small molecules, nanoparticles, or other targeting structures are being investigated due to their ability to provide radiation with shorter pathlengths (mm range)[11]. Systemic administration of these radionuclides also results in exposure to normal tissues and organs, therefore a more direct delivery system, such as convection enhanced delivery (CED)[12], is necessary.

CED relies on bulk flow, or a hydrostatic pressure gradient, to distribute infusate through the interstitial spaces of the brain tissue so both small and large molecular weight substances can be distributed homogeneously over clinically relevant volumes. Because infusate is delivered into the brain parenchyma via a catheter, the blood brain barrier is bypassed, and specific regions can be targeted for therapeutic drug delivery[13]. A number of clinical trials have been conducted using CED primarily with chemotherapeutics or targeted toxins. The first randomized, Phase III evaluation of an agent administered via CED was cintredekin besudotox versus Gliadel wafers in the PRECISE trial[14]. While negative, a retrospective analysis of the expected drug distribution based on catheter positioning data revealed that only 49.8% of catheters met all positioning criteria and overall survival ($p = 0.006$) was higher for investigators considered experienced after adjusting for patient age and Karnofsky Performance Scale score[15]. However, since that time a number of technological improvements have been implemented including improved catheter design, neuro-navigation, and in silico planning[16–18].

Liposomes are spherical, self-assembling vesicles made up of one or more naturally occurring lipid bilayers in a central compartment, which makes them ideal candidates as delivery vehicles for small molecules, proteins, nucleic acids, and imaging agents for therapeutic and diagnostic use[19–21]. Although nanoliposomes were previously shown to be amenable to loading of diagnostic imaging agents, including $^{99m}Tc$, $^{67}Ga$, and $^{111}In$[20], the mechanisms of labeling were not transferrable to all isotopes. We previously demonstrated the ability to load nanoliposomes with rhenium-186 ($^{186}Re$, half-life ~89.2 h) using a custom lipophilic molecule, N,N-bis(2-mercaptoethyl)-N',N'-diethylethylene diamine (BMEDA)[22–25] and encapsulation results in improved bioavailability and distribution of chelated-$^{186}Re$ in tumor tissue[26]. The final product—Rhenium ($^{186}Re$) Obisbemeda ($^{186}RNL$)—can be applied to both nuclear imaging and targeted radionuclide therapy.

$^{186}Re$ is a reactor produced isotope in the same chemical family as $^{99m}Tc$, the most used isotope for diagnostic imaging in nuclear medicine. $^{186}Re$ is not taken up by bone and is readily cleared by the kidneys. $^{186}Re$ is a beta emitter which exerts dose dependent therapeutic effects by inducing single strand DNA breaks[27]. Additionally, every 10$^{th}$ isotope decay also produces a 137 keV gamma photon for quantitative imaging of in vivo distribution on standard nuclear imaging equipment available in routine medical practice. $^{186}Re$ was preferentially chosen over $^{188}Re$, which is generator produced, due to its favorable mean path length of 1.8 mm (compared to $^{188}Re$'s 3.5 mm) to better avoid healthy tissue penetration for lower normal tissue toxicity[28].

In orthotopic xenograft models of glioblastoma we observed high in vivo stability of $^{186}RNL$ characterized by slow blood clearance and gradually increasing spleen accumulation[23]. Furthermore, >100 Gy of $^{186}RNL$ was shown to eradicate grafted tumors and prolong overall survival, without clinical or microscopic evidence of toxicity[23]. A preclinical laboratory study conducted in accordance with the United States Food and Drug Administration (FDA) Good Laboratory Practice Regulations was subsequently conducted to assess overall toxicity and evaluate dosimetry of a single dose administration of $^{186}RNL$ (unpublished data; Brenner A, Floyd J, Bao A, Phillips WT, & Goins B). Intracranial administration of 1, 3.5, or 6 mCi $^{186}RNL$ or control article produced no significant test article-related pathologic changes systemically or in the brains of female beagle dogs at 24 h or 14 days. Macroscopic and microscopic changes seen were judged to be a result of the dosing procedure. Based on these data, the no adverse effect limit as related to brain pathology was determined to be 6 mCi $^{186}RNL$ as a single infusion when assessed at 24 h and 14 days.

Prior to study start we searched PubMed from any time up to December 1, 2013 for clinical therapeutic studies using $^{186}Re$ to treat glioma using the search terms "($^{186}Re$[ALL FIELDS] AND glioma[ALL FIELDS]) and (rhenium-186[ALL FIELDS] AND glioma[ALL FIELDS]). The search gave four publications[29–32]. Intramedullary cystic spinal cord pilocytic astrocytoma was managed with minor side effects using $^{186}Re$ intracavitary irradiation, with stabilization of the cyst and neurological deficit improvement[29]. Combination treatment for pilocytic astrocytoma using $^{186}Re$ delivered 400 Gy to the cyst wall and resulted in progressive cyst disappearance and mural nodule retraction[32]. A fibrin glue of $^{188}Re$ and $^{186}Re$ bound in microspheres was used post-tumor resection in a 9L-glioblastoma rat model. 60% of treated animals survived 36 days, compared to control animals ($17 \pm 3$ days)[30]. Lastly, intracavitary $^{186}Re$ application was used in six cases of cystic craniopharyngiomas but abandoned because of cyst recurrence and leakage[31].

Following preclinical studies, we initiated ReSPECT-GBM, a multicenter, sequential cohort, open-label, volume and dose escalation study of the safety, tolerability, and distribution of a single dose of $^{186}RNL$ given by CED for recurrent glioma. The primary objective was to determine the maximum tolerated dose (MTD) of $^{186}RNL$ by CED in patients with recurrent glioma. Secondary objectives included the assessment of the safety and tolerability of a single dose $^{186}RNL$, the dose distribution of $^{186}RNL$ by imaging, the overall response rate (ORR), disease-specific progression-free survival (PFS), and OS. A graphical summary of the trial design and its outcomes in included in Supplementary Fig. 4.

## Results

### Demographics

Patients were enrolled in the study between March 5, 2015 and April 22, 2021, with 3 screen failures due to finding a recently developed lesion on the pretreatment MRI, tumor volume size, and a decision by the patient to not be treated (Fig. 1). The study used a modified 3 + 3 dose escalation[33], with increased in total radioactivity by doubling from 1 mCi in Cohort 1 to 8 mCi in Cohort 4, followed by a 67% increase in Cohorts 5 and 6. All administered doses were determined prior to study start, with dose escalation to a subsequent Cohort following confirmation by the Data Safety Monitoring Board (DSMB), after a review of all safety data. The administered dose range was 1 mCi in a volume of 0.66 mL through 22.3 mCi in 8.80 mL (Table 1).

Among the 21 patients treated in Cohorts 1–6, the mean age was $53.1 \pm 10$ (35, 69), 66.7% were male, and 90.5% were white. Using the 2016 WHO classification[34], two were Grade 3 and 19 were Grade 4. 13 (62%) patients were first recurrence, 7 (33%) second recurrence, and 1 (5%) third recurrence or later. For prior treatments reported, 20 (95.2%) received temozolomide, 21 (100%) received radiation, 20 (95.2%) received surgery, and 6 (28.6%) received prior bevacizumab. Additional prognostic features included *IDH1* status and MGMT methylation status (Table 1). Tumor locations were frontal in 5 (24%), parietal in 3 (14%), temporal in 6 (29%), occipital in 2 (9%), and overlapping lobes in 5 (24%).

### Safety

Safety was determined by utilizing the grading scale of the CTCAE v4.0[35] to assess the severity of adverse events (AEs). There were no dose limiting toxicities (DLTs) found. Most AEs (Table 2) were mild (71.7%) to moderate (21.4%), with 6.9% graded as severe (Table 3). Most AEs were not attributable to study drug (68.6% unrelated, 17.6% unlikely, 12.6% possible) (Table 3). Only one AE of scalp discomfort (grade 1) was considered related to the CED catheter placement (0.6%, definite); no AEs were determined to be related to $^{186}RNL$ (Table 3). Most AEs reported (>5%) were fatigue (57.1%) and headache (47.6%)

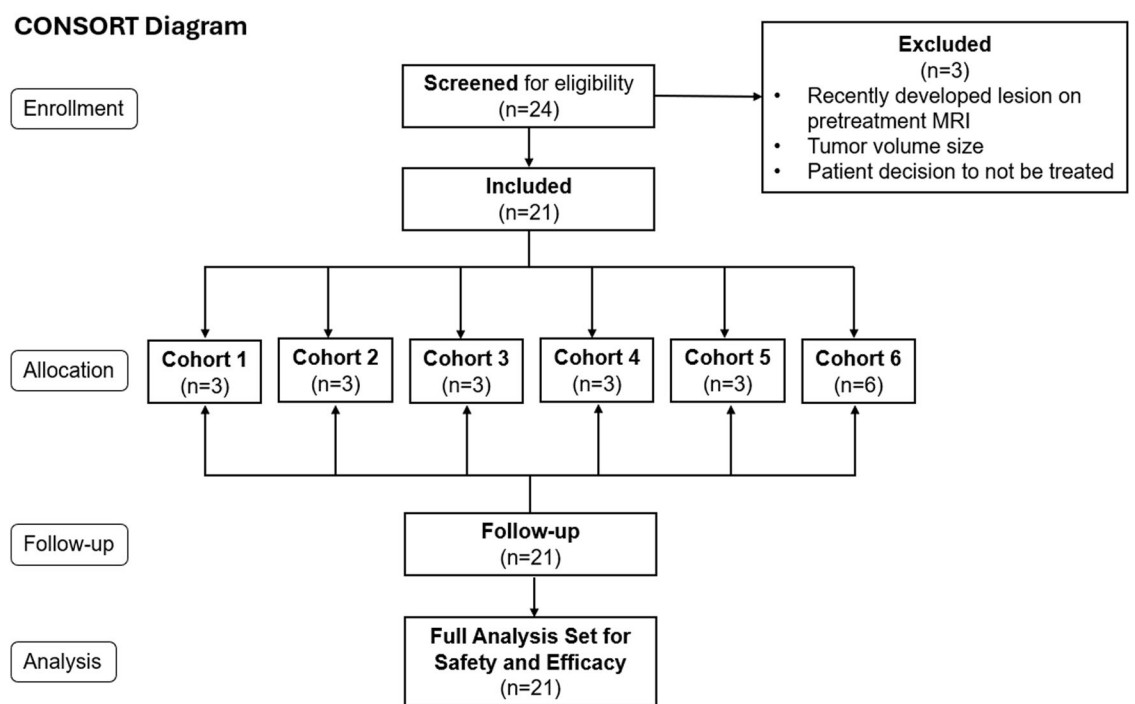

**Fig. 1 | CONSORT diagram.** Twenty-four patients were screened for eligibility with 21 included in the study. Reason for exclusion and allocation per cohort are as detailed.

## Table 1 | Per cohort dose escalation plan, demographics, and genetic characteristics

| Dosing Variables | | Cohort 1 | Cohort 2 | Cohort 3 | Cohort 4 | Cohort 5 | Cohort 6 | |
|---|---|---|---|---|---|---|---|---|
| Infused volume of $^{186}$RNL (mL) | | 0.66 | 1.32 | 2.64 | 5.28 | 5.28 | 8.8 | |
| Administered dose of $^{186}$RNL (mCi) | | 1 | 2 | 4 | 8 | 13.4 | 22.3 | |
| Concentration of $^{186}$RNL (mCi/mL) | | 1.5 | 1.5 | 1.5 | 1.5 | 2.5 | 2.5 | |
| Number of Patients | | 3 | 3 | 3 | 3 | 3 | 6 | |
| **Characteristics** | | **Cohort 1** | **Cohort 2** | **Cohort 3** | **Cohort 4** | **Cohort 5** | **Cohort 6** | **Total** |
| Age mean ± SD (Range) | | 52 ± 12.1 (39, 63) | 60 ± 10.1 (49, 69) | 49.3 ± 15 (35, 65) | 53 ± 15.1 (36, 65) | 51 ± 8.5 (42, 59) | 53.2 ± 7 (41, 60) | 53.1 ± 10 (35, 69) |
| Tumor volume mean ± SD (Range) | | 1.8 ± 1.5 (0.9, 3.5) | 4.2 ± 3.7 (2, 8.5) | 13 ± 9.1 (2.5, 18.4) | 8 ± 6.5 (2.4, 15.1) | 11.7 ± 6.8 (3.9, 16) | 9.5 ± 10.1 (1, 22.8) | 8.3 ± 7.7 (0.9, 22.8) |
| Male n (%) | | 3 (100) | 2 (66.7) | 2 (66.7) | 2 (66.7) | 2 (66.7) | 3 (50) | 14 (66.7) |
| White n (%) | | 3 (100) | 3 (100) | 3 (100) | 2 (66.7) | 3 (100) | 5 (83.3) | 19 (90.5) |
| ECOG Performancen (%) | 0 | 1 (33.3) | 1 (33.3) | 3 (100) | 1 (33.3) | 3 (100) | 6 (100) | 15 (71.4) |
| | 1 | 1 (33.3) | 1 (33.3) | 0 (0) | 1 (33.3) | 0 (0) | 0 (0) | 3 (14.3) |
| | 2 | 1 (33.3) | 1 (33.3) | 0 (0) | 1 (33.3) | 0 (0) | 0 (0) | 3 (14.3) |
| IDH1 Mutation Status n (%) | Mutated | 0 (0) | 0 (0) | 1 (33.3) | 1 (33.3) | 0 (0) | 0 (0) | 2 (9.5) |
| | Wild type | 2 (66.7) | 1 (33.3) | 1 (33.3) | 2 (66.7) | 3 (100) | 6 (100) | 15 (71.4) |
| | QNS | 1 (33.3) | 1 (33.3) | 0 (0) | 0 (0) | 0 (0) | 0 (0) | 2 (9.5) |
| | NOS | 0 (0) | 1 (33.3) | 1 (33.3) | 0 (0) | 0 (0) | 0 (0) | 2 (9.5) |
| MGMT methylation status n (%) | Methylated | 0 (0) | 1 (33.3) | 1 (33.3) | 0 (0) | 1 (33.3) | 1 (16.7) | 4 (19) |
| | Unmethylated | 2 (66.7) | 1 (33.3) | 2 (66.7) | 3 (100) | 2 (66.7) | 4 (66.7) | 14 (66.7) |
| | QNS | 1 (33.3) | 1 (33.3) | 0 (0) | 0 (0) | 0 (0) | 1 (16.7) | 3 (14.3) |
| Stage at screening (%) | III | 1 (4.8) | 0 (0) | 0 (0) | 0 (0) | 1 (4.8) | 0 (0) | 2 (9.5) |
| | IV | 2 (9.5) | 3 (14.3) | 3 (14.3) | 3 (14.3) | 2 (9.5) | 6 (28.6) | 19 (90.5) |

Cohort 6 patients 4–6 had an increase in maximum flow rate to 20 microliters per minute, but the same volume and dose as Cohort 6 patients 1–3.

(Table 2) and resolved without treatment (Supplementary Table 1). Of serious adverse events (SAEs) reported (10) (Supplementary Tables 2 and 3), all were grade 3 or less. There were no instances of post-operative intracranial hemorrhages from the SoC biopsy site or from catheter placements. Only one SAE (vasogenic cerebral edema) was considered possible to $^{186}$RNL due to overly rapid glucocorticoid medication taper which was promptly resolved with medication adjustment. While the primary endpoint was MTD, an MTD was not reached, and a recommendation from the DSMB was made to proceed to Phase 2 with Cohort 6 dosing parameters for tumors 20 ml or less.

**Table 2 | Adverse events experienced by at least 10% of patients (*N* = 21) with multiplicities within a patient eliminated by selecting the AE with the maximum grade**

| Adverse Event | Cohort 1 | Cohort 2 | Cohort 3 | Cohort 4 | Cohort 5 | Cohort 6 | Frequency (%) |
|---|---|---|---|---|---|---|---|
| Fatigue | 2 | 1 | 1 | 2 | 2 | 4 | 12 (57.1) |
| Headache | 1 | 2 | 1 | 1 | 1 | 4 | 10 (47.6) |
| Diarrhea | 1 | 1 | 2 | 0 | 0 | 1 | 5 (23.8) |
| Dizziness | 1 | 0 | 2 | 0 | 0 | 2 | 5 (23.8) |
| Edema cerebral | 0 | 1 | 1 | 0 | 1 | 2 | 5 (23.8) |
| Gait disturbance | 0 | 2 | 1 | 0 | 2 | 0 | 5 (23.8) |
| Seizure | 1 | 1 | 0 | 1 | 1 | 1 | 5 (23.8) |
| Alanine aminotransferase increased | 1 | 1 | 1 | 0 | 1 | 0 | 4 (19) |
| Anorexia | 0 | 0 | 0 | 2 | 1 | 0 | 3 (14.3) |
| Aspartate aminotransferase increased | 1 | 1 | 1 | 0 | 0 | 0 | 3 (14.3) |
| Constipation | 0 | 1 | 1 | 0 | 0 | 1 | 3 (14.3) |
| Dysesthesia | 0 | 0 | 0 | 1 | 1 | 1 | 3 (14.3) |
| Dysphasia | 1 | 0 | 0 | 1 | 0 | 1 | 3 (14.3) |
| Edema limbs | 0 | 1 | 0 | 2 | 0 | 0 | 3 (14.3) |
| Generalized muscle weakness | 0 | 1 | 0 | 1 | 0 | 1 | 3 (14.3) |
| Memory impairment | 0 | 0 | 0 | 0 | 0 | 3 | 3 (14.3) |
| Muscle weakness lower limb | 1 | 1 | 1 | 0 | 0 | 0 | 3 (14.3) |
| Nausea | 0 | 1 | 1 | 0 | 0 | 1 | 3 (14.3) |
| Tinnitus | 1 | 1 | 1 | 0 | 0 | 0 | 3 (14.3) |
| Vomiting | 1 | 0 | 1 | 0 | 0 | 1 | 3 (14.3) |

**Table 3 | Per Cohort grades and relation of all adverse events**

| Grade | Cohort 6 | Cohort 5 | Cohort 4 | Cohort 3 | Cohort 2 | Cohort 1 |
|---|---|---|---|---|---|---|
| 1-Mild | 24 (64.9) | 12 (70.6) | 12 (60) | 22 (84.6) | 23 (67.6) | 21 (84) |
| 2-Moderate | 10 (27) | 3 (17.6) | 4 (20) | 4 (15.4) | 9 (26.5) | 4 (16) |
| 3-Severe | 3 (8.1) | 2 (11.8) | 4 (20) | 0 (0) | 2 (5.9) | 0 (0) |
| *Totals* | 37 | 17 | 20 | 26 | 34 | 25 |
| **Relation** | **Cohort 6** | **Cohort 5** | **Cohort 4** | **Cohort 3** | **Cohort 2** | **Cohort 1** |
| Definite | 0 (0) | 0 (0) | 0 (0) | 0 (0) | 0 (0) | 1 (4) |
| Possible | 12 (32.4) | 3 (17.6) | 3 (15) | 0 (0) | 0 (0) | 2 (8) |
| Probable | 1 (2.7) | 0 (0) | 0 (0) | 0 (0) | 0 (0) | 0 (0) |
| Unlikely | 14 (37.8) | 7 (41.2) | 3 (15) | 1 (3.8) | 0 (0) | 3 (12) |
| Unrelated | 10 (27) | 7 (41.2) | 14 (70) | 25 (96.2) | 34 (100) | 19 (76) |
| *Totals* | 37 | 17 | 20 | 26 | 34 | 25 |

## Volume of distribution and dosimetry

Whole-body planar and SPECT/CT imaging at serial time points were performed to determine the volume of $^{186}$RNL distributed in the brain and in the tumor, percentage of tumor volume treated (TVT), radiation absorbed dose to the tumor, and whole body normal organ doses.

Early cohorts (1–3) explored safety across all variables, with one catheter used and $^{186}$RNL volumes not exceeding 3 mL; further cohorts (4–6) expanded these parameters. The mean volume of distribution (mVd) across all cohorts was 44.19 mL and increased over each cohort (Fig. 2). The mVd for Cohort 6 was 65.57 mL and had the highest median Vd (69.25 mL) compared to all other cohorts. The percentage of TVT (at 120 h post-treatment) ranged from 7.6% to 100%, with 12/21 tumors receiving ≥70% coverage. Persistence of activity at the site of administration was noted at all imaging timepoints confirming excellent retention (Fig. 3).

The majority of patients (47.6%) had 1 catheter (Supplementary Table 7). Three patients had catheter "failure," where it was observed that one catheter did not have the same flow as the others due to a blockage ('kink') in the delivery tubing or drug was being lost to an area of lower resistance; in these cases, all patients received the full amount of administered dose by diverting the volume of the failed catheter to another catheter in use.

The radiation absorbed dose to the tumor ranged from 8.9 Gy to 739.5 Gy. Absorbed dose per cohort is shown in Fig. 2. Radiation absorbed dose to the tumor correlated positively with the percent of tumor covered ($r = 0.79$, $p < 0.001$) (Supplementary Fig. 1), the volume of $^{186}$RNL infused ($r = 0.37$, $p = 0.10$), the Vd ($r = 0.15$, $p = 0.51$), and the number of catheters ($r = 0.16$, $p = 0.48$). It correlated negatively with age ($r = -0.32$, $p = 0.17$), baseline ECOG status ($r = -0.58$, $p = 0.006$), and baseline tumor volume ($r = -0.42$, $p = 0.060$). All 12 patients with ≥70% tumor coverage had ≥100 Gy absorbed dose.

Urine collected from 1–24 h to 24–48 h showed the highest amounts of radioactivity in the first interval, with a percent injected dose (%ID) of 10.06% ± 10.94 (mean ± SD) and a range of 0.65–44.50%. The absorbed dose in blood was measured over time; at 192 h, the mean absorbed dose was 3.180 ± 4.87 cGy.

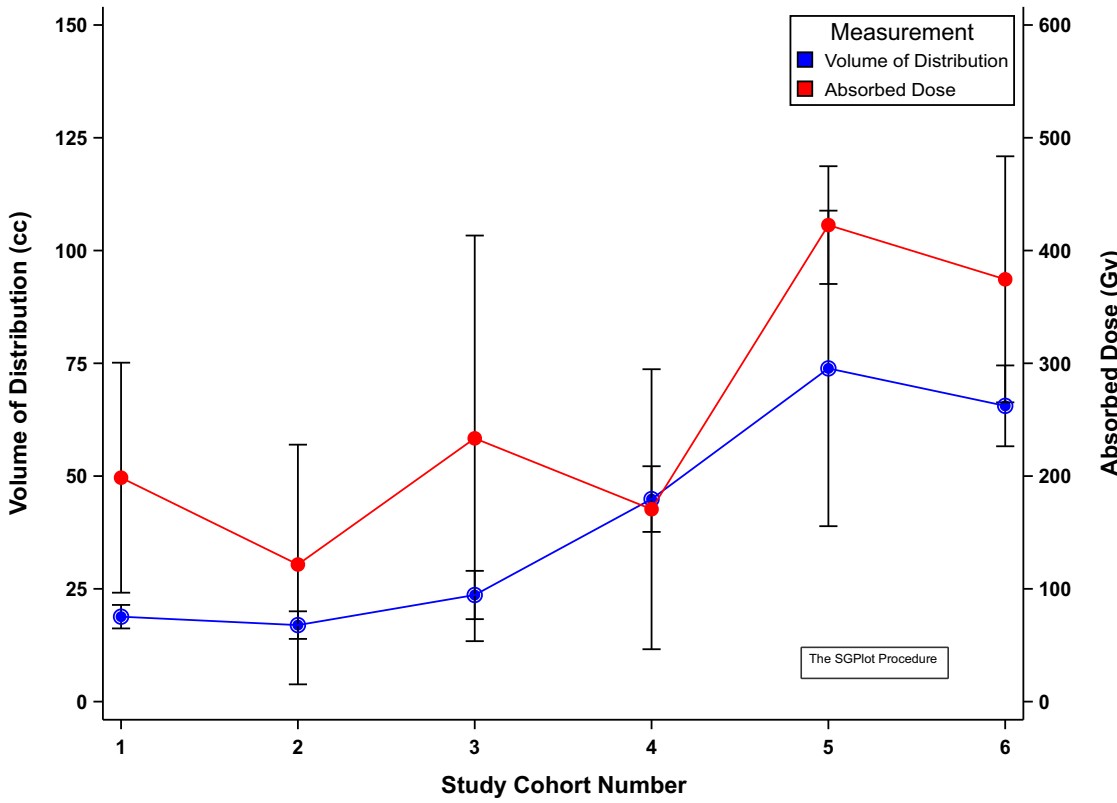

**Fig. 2 | Mean volume of distribution and absorbed dose by cohort.** The volume of distribution (left y-axis) and corresponding absorbed dose (right y-axis) are shown for each cohort (x-axis) with standard error bars showing an increase in both distribution and absorbed dose through cohort 5, without significant increase in cohort 6.

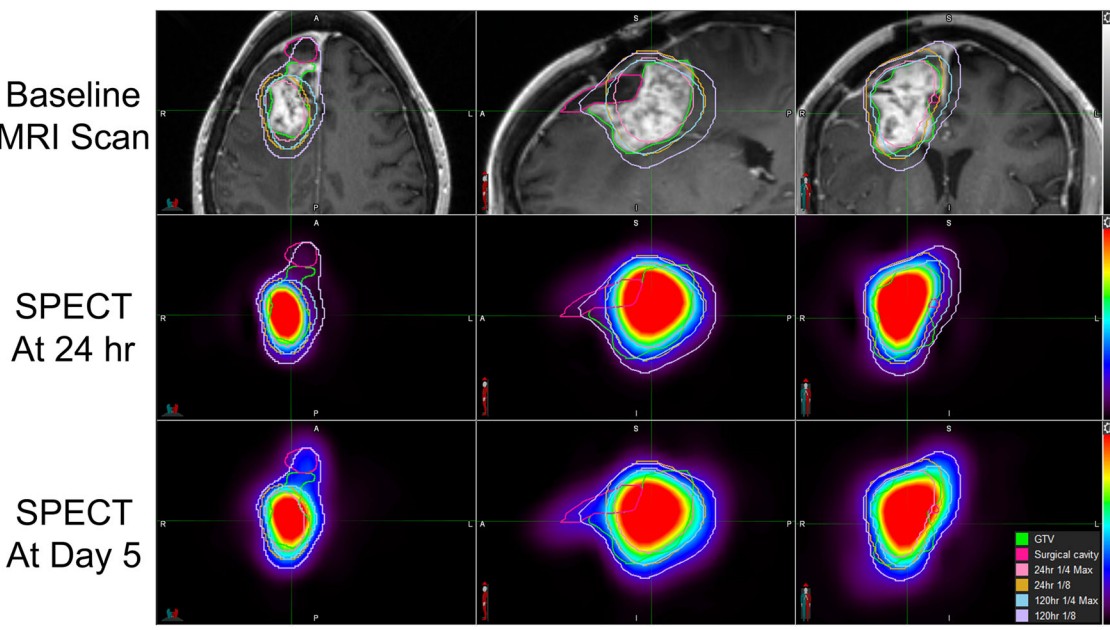

**Fig. 3 | MRI and SPECT/CT scan showing distribution and retention of rhenium obisbemeda through Day 5.** Top row: T1 contrast enhanced MRI in the axial, sagittal, and coronal plains with overlayed SPECT outlines for day 5 SPECT activity. Middle row: SPECT images from corresponding planes at 24 h. Bottom row: SPECT images from corresponding planes at 5 days showing persistent activity.

Whole body planar imaging showed mean normalized organ absorbed radiation doses were highest in the liver ($1.99 \pm 1.41$ cGy/mCi), spleen ($3.00 \pm 4.33$ cGy/mCi), and urinary bladder wall ($2.46 \pm 2.71$ cGy/mCi) (Fig. 4, Supplementary Fig. 2). Because of its smaller size, the spleen is anticipated to be the critical organ for dosimetry calculations; the absorbed dose was well within acceptable absorbed doses for these organs. No other organs showed clinically significant uptake of [186]RNL, besides the brain, which included the absorbed dose to the tumor.

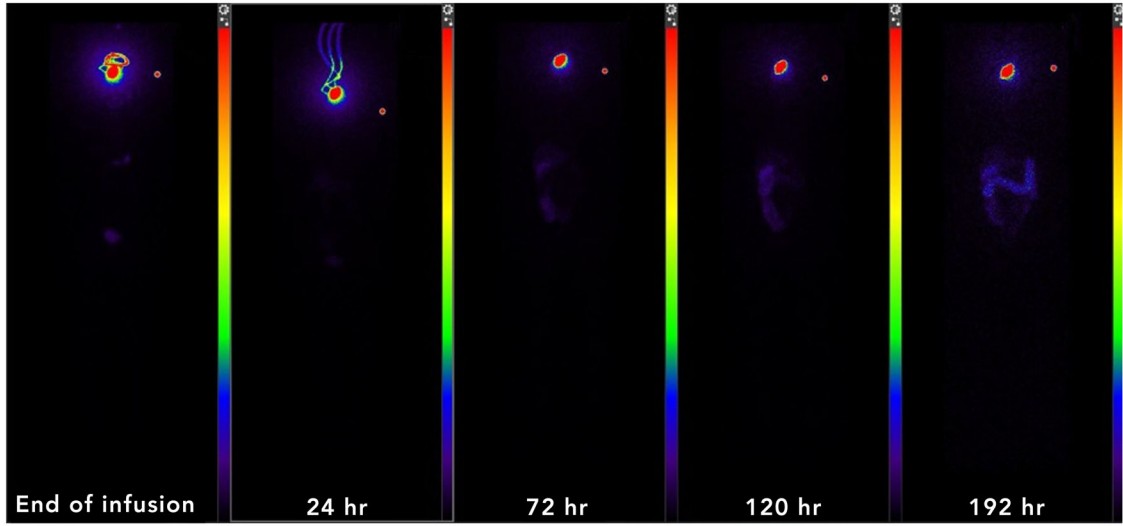

**Fig. 4 | Whole body planar imaging demonstrating retention within the tumor and lack of significant activity in the remainder of the body.** Patient example of whole body planar imaging demonstrating retention within the tumor and catheters without significant activity in the remainder of the body at various time points as labeled.

### Table 4 | Best response by Cohort

| Best Response | Cohort 6 | Cohort 5 | Cohort 4 | Cohort 3 | Cohort 2 | Cohort 1 | Total Frequency (%) |
|---|---|---|---|---|---|---|---|
| PR | 1 (16.7) | 0 (0) | 0 (0) | 0 (0) | 0 (0) | 0 (0) | 1 (4.8) |
| SD | 3 (50) | 2 (66.7) | 3 (100) | 1 (33.3) | 1 (33.3) | 2 (66.7) | 12 (57.1) |
| PD | 2 (33.3) | 1 (33.3) | 0 (0) | 2 (66.7) | 2 (66.7) | 1 (33.3) | 8 (38.1) |

*PR* partial response, *SD* stable disease, *PD* progressive disease.

### Efficacy

ORR was determined by RANO criteria[36] and defined as the proportion of patients with complete or partial response. PFS was defined as the time from dosing to documented disease progression as determined by the investigators, clinical progression in the absence of MRI determination, or death from any cause, whichever occurred first. OS was defined as the time between the first day of study treatment until death from any cause. For ORR, 12 patients (57.1%) exhibited stable disease (SD), one (4.8%) partial response (PR), and eight progressive disease (PD) (38.1%) (Table 4).

The median PFS (mPFS) was 4.0 months (95% CI 2.0–6.0 months) with one patient progression-free >11 months (Supplementary Fig. 3a). The mOS across all cohorts 1 to 6 ($n = 21$) was 11.0 months (95% CI 5.0–17.0 months) with one patient remaining alive 29 months after treatment (Fig. 5a). A total of 12/21 patients received ≥100 Gy radiation absorbed dose (Supplementary Table 8).

### Post-hoc analysis

Several post-hoc analyses not predefined in the Phase 1 study protocol were also performed. Given the findings from our preclinical studies identifying 100 Gy as a threshold for survival benefit[23], patients were also dichotomized by absorbed dose of 100 Gy. After dichotomizing by absorbed dose, a significant difference was observed with mPFS, with 2.0 months (95% CI 1.0–4.0 months) for <100 Gy and 6.0 months for ≥100 Gy (95% CI 3.0–8.0 months) (Supplementary Fig. 3b). After adjustment for age, baseline ECOG status, baseline volume administered, and baseline tumor volume, PFS increased by 15% for each 10% increase in the percentage of tumor covered (fold change=1.49, 95% CI 1.063, 1.242) and by 19% for each 100 Gy increased in the absorbed dose (fold change = 1.187, 95% CI 1.04, 1.356) (Supplementary Table 4).

The number of catheters significantly correlated with Vd ($r = 0.67$, $p < 0.001$) and $^{186}$RNL volume infused ($r = 0.69$, $p < 0.001$), and not with tumor coverage (%) ($r = 0.39$, p = 0.08), age ($r = -0.04$, $p = 0.88$), baseline ECOG status ($r = -0.41$, $p = 0.07$), or absorbed dose ($r = 0.16$, $p = 0.48$).

A significantly longer mOS was observed in those whose tumors received >100 Gy ($p < 0.001$). The mOS was 17.0 months (95% CI 8.0–35 months) for >100 Gy ($n = 12$) compared to 6.0 months (95% CI 1.0–11.0 months) for <100 Gy ($n = 9$) (Fig. 5b).

After adjustment for age, baseline ECOG status, baseline volume administered, and baseline tumor volume, OS for all cohorts increased by 27% for each 10% increase in the percentage of tumor covered (fold change 1.274 95% CI 1.209–1.343, $p < 0.001$) and by 31% for each 100 Gy increase in the absorbed dose (fold change 1.312, 95% CI 1.124–1.532, $p < 0.001$) (Supplementary Table 5).

### Discussion

We report the safety, pharmacokinetics, and preliminary efficacy of a novel radiotherapeutic, $^{186}$RNL, in a Phase 1 study for the treatment of recurrent glioma. Directly targeting tumors using CED, we were able to safely deliver up to 8.8 mL of 22.3 mCi administered dose using up to 4 catheters, delivering much higher radiation doses than can be delivered with EBRT. In this initial analysis, the MTD was not reached, and OS was shown to closely correlate with both percent tumor volume coverage by the drug and the radiation absorbed dose to the tumor.

EBRT is an essential component of glioblastoma therapy. It is tumoricidal based on the generation of free radicals but injury to surrounding normal tissues in the beam path significantly limits dose. Brachytherapy is an alternate form of radiation delivery. The radiation source is placed adjacent to the treatment area, sparing normal tissues. Ideal isotopes in CNS brachytherapy, as demonstrated herein with

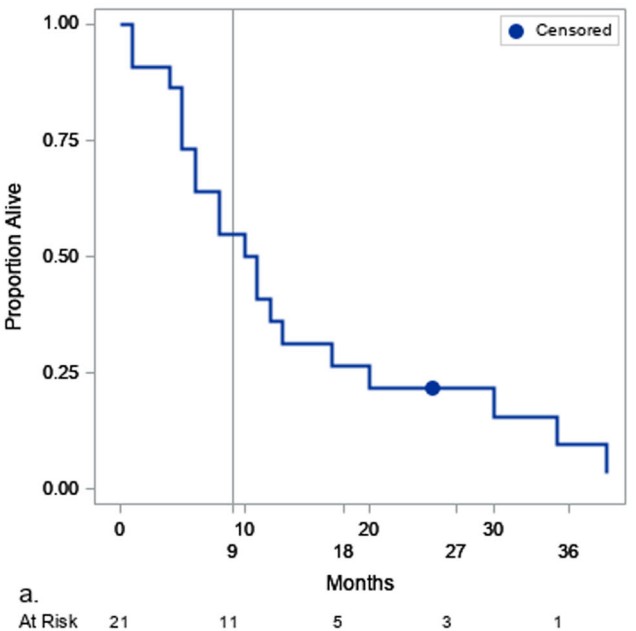

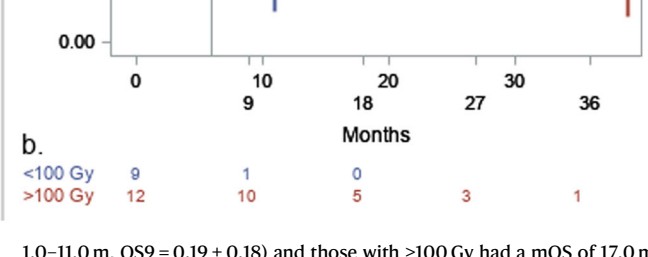

**Fig. 5 | Kaplan-Meier of overall survival. a** For all patients, the mOS was 11.0 m (95% CI 5.0–17.0 m, OS9 = 0.55 ± 0.11). **b** When dichotomized by absorbed dose, patients who received <100 Gy had a mOS of 6.0 m (95% CI 1.0–11.0 m, OS9 = 0.19 ± 0.18) and those with ≥100 Gy had a mOS of 17.0 m (95% CI 8.0–35.0 m, OS9 = 0.84 ± 0.11). Source data are provided as a Source Data file.

[186]Re, should have several features including short path length, low dose rate, high energy output, and a means to image and compute dose. Furthermore, the isotope should address both enhancing and non-enhancing tumor and remain in place throughout the decay cycle of the isotope to minimize systemic exposure.

As noted, the MTD was not reached, and no DLTs were observed. Most AEs were mild or moderate in grade. Only one SAE was deemed possibly related to [186]RNL, likely due to too rapid a corticosteroid taper. When OS and PFS were modest, these patients were notable in poor percent tumor coverage and low radiation absorbed dose. An additional confounder in progression assessment is increased vascular permeability from cytotoxic therapies, which has been shown to result in increased contrast enhancement in the context of therapeutic benefit, phenomena known as "pseudoprogression"[36]. This was observed in several patients on this study who showed asymptomatic increased enhancement at Day 28 post-treatment with significant stability thereafter. Operative excision in one case which was radiographically indeterminate showed tumor necrosis and vascular changes without residual tumor and no signs of radiation-induced necrosis or vasculopathy.

For safety considerations, infused drug volume, total radioactivity, dose concentration, infusion flow rates, and number of CED catheters began at conservative levels. As favorable safety and tolerability was observed, these parameters increased over cohorts. Generally, in constant tumor sizes, increasing these parameters correlated with higher absorbed doses of radiation to the tumors and increased tumor coverage, though tumors sizes overlapped intra and intercohort. Following interim analysis of Cohort 6, we observed a plateau from Cohort 5 to Cohort 6 in both dose distribution and absorbed dose. As a result of this finding, the recommendation of the DSMB was to expand Cohort 6 to confirm tolerability of 22.3 mCi administered dose in 8.8 mL infusate volume as the Phase 2 recommended dose (RP2D) in patients with one recurrence and tumor sizes of 20 cm³ or less, to target both bulk disease and adjacent microscopic disease. Once completed, and without the MTD being reached, all cohort data was analyzed with recommendation from the DSMB to proceed with Phase 2 utilizing this dose.

[186]Re emits both beta and gamma particles. The gamma emission enables real-time imaging and precise calculation of absorbed radiation dose to the tumor. In this study the maximum absorbed dose was ~740 Gy. Both preclinical dose finding studies and statistical modeling of the clinical data with long rank testing suggested a dose effect of greater than 100 Gy. When this dose threshold was applied, a significant marginal correlation was observed between dose, both in its binary (<100 Gy, >100 Gy) and continuous form, and OS. A similar marginal correlation was also observed between percent tumor coverage and OS. Furthermore, these correlations remained statistically significant after adjustment for baseline factors in multivariate models. While further assessment of the data using a combination of perfusion and delayed contrast imaging is ongoing, we have selected a primary endpoint of OS for our Phase 2 study, currently underway.

Although we found a correlation between the number of catheters, volume of distribution, and the administered volume, we did not observe a correlation between number of catheters and percent tumor coverage. There are a number of confounders that may impact correlation between number of catheters and percent tumor coverage, particularly that treatment planning for larger tumors likely would drive a decision to place a higher number of catheters, but that the inherent challenges of larger tumors would make coverage less successful. Coverage of greater than 70% appeared to be associated with benefit (Supplementary Fig. 1b), but this is likely a factor of the fixed thresholding method used relative to maximum voxel radioactivity (count) of each SPECT image, and that treatment effects extend beyond this region which may be better discriminated with 3D dosimetry. Similarly, a correlation between administered dose and survival was not observed, and likely reflects the decision to match tumor volume in the inclusion criteria to the volume of administration while holding the concentration constant through the first four cohorts.

Several limitations were present in this study which will require further evaluation. First, since this is a first-in-human dose escalation study, including varying doses, administration rates, number of catheters, and tumor sizes, any results should be taken as preliminary and will require further validation. From a dose distribution perspective, SPECT as an imaging modality to be used for dosimetry and

absorbed dose quantification has inherent technological limitations when addressing CNS tumors based on camera resolution, voxel size, and brain anatomy. Further, distribution was estimated based on activity rather than isodose lines. We have since begun performing three-dimensional dosimetry which can define treatment volumes based upon absorbed dose isodose lines and will better define biologically relevant distribution. CED also is technically challenging and can be a source of treatment failure even with an efficacious therapeutic. We recognize the infrastructure required, including sophisticated, cost-intensive neuronavigation software and hardware. At present, this technology from companies such as Brainlab or Stryker are currently available in most hospitals with neuro-oncology expertise, and routinely used. For this study and our ongoing Phase 2 trial, treatment planning is performed centrally with participation of the local neurosurgeon. As reported, following early cohorts, multiple catheters were used whenever possible to maximize distribution, minimize convection time, improve coverage of irregular shaped tumors, and to provide redundancy in case of catheter failure. The ability to visualize the delivery of the therapeutic directly at the time of administration is a unique advantage of CED with a radiopharmaceutical, and further analysis of the entire data set is ongoing and will be reported. Finally, determining response can be challenging in CNS tumors, particularly with radiation which can result in pseudo-progression. To better characterize response, we are performing advanced imaging modalities including Delta T1 subtraction mapping, DSC perfusion, and delayed contrast mapping (TRAMs). In preliminary parametric analyses we observed a statistically significant difference in repeated pair measurements with untreated tumor volume significantly increased relative to treated tumor volume ($p < 0.0001$) suggesting that progression arises outside the treated volume.

The combination of a novel nanoliposome radiotherapeutic delivered by CED, facilitated by neuronavigational tools, catheter design, and imaging solutions can successfully and safely provide high absorbed radiation doses to tumors with minimal toxicity and potential survival benefit.

## Methods

### Eligibility

This human study was performed in accordance with the Declaration of Helsinki and was approved by the Western Institutional Review Board. The study's clinical trial registration number is NCT01906385, registered with https://clinicaltrials.gov/study/NCT01906385, and the date of first public posting was July 24, 2013. Participant registration took place from Mar-2015 to Apr-2021. Patients were enrolled at two study sites: UT Heath San Antonio and UT Southwestern Medical Center. Eligible participants were at least 18 years of age, able to provide written consent and had histologically confirmed recurrent WHO Grade 3 or 4 glioma with an enhancing tumor volume within the treatment field volume in the respective cohort, based on a the most recent (within 35 days) magnetic resonance imaging (MRI) assessment. The 2016 WHO classification[34] was utilized throughout the study. Only two patients (one each from Cohorts 4 and 5) had the IDH1 mutation (Table 1), making them not glioblastoma under the current 2021 WHO classification[2], which did not meaningfully alter the findings of this study and were therefore included in the analysis. Patients were not limited by their number of recurrences. Recurrences were defined as progression by RANO criteria or other clinically accepted neuro-oncology evaluation. Patients were further limited to having completed standard treatment options with known survival benefit for any recurrence (e.g., surgery, temozolomide, radiation, and tumor treating fields), but were allowed on study if medically unable or unwilling to follow standard treatment options for any recurrence. Additionally, the protocol was modified after Cohort 4 to exclude patients with Grade 3 glioma or prior bevacizumab treatment, as some of the early patients with prior bevacizumab therapy showed poor [186]RNL

convection to the tumor, lower tumor coverage, and lower tumor absorbed dose. Progression was determined by Radiographic Assessment in Neuro-Oncology (RANO) criteria[36] following standard treatment.

Additional eligibility requirements included an ECOG performance status of 0–2, acceptable liver and renal function, stable hematologic status without transfusion, stable anti-epileptic medication and seizures, and stable or decreasing corticosteroids to control cerebral edema. Women of childbearing age were required to have a negative serum pregnancy test and use effective means of contraception from entry into the study through 6 months after the last dose.

Participants were excluded from the study if they had multifocal progression, involvement of the leptomeninges, infratentorial disease, evidence of acute intracranial hemorrhage, AEs grade >1 by the National Cancer Institute (NCI) Common Terminology Criteria for Adverse Events (CTCAE) v4.0[35] from any medications administered prior to study, a serious intercurrent illness, inherited bleeding diathesis or coagulopathy, or received any non-standard radiation therapy.

No sentinel patients were included in this study. The first patient dosed was under the FDA-approved protocol in Cohort 1. The study protocol is available in the Supplementary Information file.

### Sex and gender

Sex and gender, as determined based on self-reporting, was not a factor in patient inclusion criteria, nor were there sufficient numbers of patients of either gender to make a meaningful sex- and gender-based conclusions in the study; as such, sex- and gender-based analyses have not been reported here.

### Treatment

Once patients were consented, they underwent treatment planning MRI within 7 days before drug infusion to evaluate tumor characteristics, including location, structure, shape, and dimension. Based on the treatment planning MRI, patients were further selected for study status. If the tumor had progressed beyond the screening criteria, they were considered a screen failure. Once confirmed to be appropriate for CED, stereotactic treatment planning was centralized and performed by the Principal Investigator (PI) together with the local neurosurgeon. The tumor volume was calculated by the use of iPlan Flow, as was all in silico treatment planning and convection simulation, as has been previously described[18]. Briefly, using the iPlan software, the PI would determine the number of catheters and their trajectories, based on the size and location of the tumor, in accordance with pre-set parameters noted below. Early cohorts minimized the number of catheters for safety and to characterize the distribution; once this was determined, the number of catheters used (1–4) was based on planning to allow for suitable coverage of the enhancing tumor with surrounding T2 FLAIR abnormality. Immediately following a SoC stereotactic biopsy procedure for confirmation of disease progression, BrainLab Flexible CED catheter(s) (7.5 mm tip) were placed, with at least one catheter proceeding along the same needle track as the SoC biopsy. Catheter placement was optimized to avoid ependymal surfaces by >0.5 cm as they offer no resistance to fluid flow and 2.0 cm from the closest sulcus, fissure, resection cavity, or cortical surface. Distal placement of the infusing tip within the tumor was also critical to avoid the pressure gradient from the tumor core to periphery to adjacent brain. Postoperative head CT was performed to evaluate for hematoma or pneumocephalus.

To allow for fibrin and clot deposition to occur, [186]RNL infusion was performed ~24 h following catheter placement. [186]RNL was manufactured as previously described[23]. Super-saturated potassium iodide (SSKI, 600 mg) was administered by mouth with water or juice prior to infusion. The infusion rate started at 1 μl/min and stepped to 20 μl/min (Table S6) using a syringe pump (Medfusion 3500 and 4000, Adepto

Medical, Kansas City, MO). The total time of infusion was a function of flow rate, number of catheters used, and total volume infused (Supplementary Tables 6 and 7, Table 1). During infusion, planar and tomographic images were collected using a dual-detector SPECT/CT camera. A sealed vial with known [186]RNL was positioned next to the patient at each time of image acquisition for radioactivity quantification. Dynamic images were acquired via real-time persistence scope for evaluation of focal accumulation of activity at the assumed tip of the catheter(s). When activity was observed to accumulate focally, that time was designated as the beginning of the planned therapeutic volume infusion to correct for dead space in the catheter line. Catheters deemed unsatisfactory (backflow along the catheter or spillage into adjacent CSF space) were stopped and the remaining volume switched to the remaining catheters.

To minimize backflow concerns[37], we utilized the Brainlab catheter, which is both flexible and has a "step design" at the tip, avoided resection cavities and pial surfaces, and employed a slow ramping of infusion rates (Supplementary Table 6). Furthermore, backflow was monitored during infusion using planar imaging as mentioned above, allowing changes to be made in real-time to ensure the planned administered dose was delivered in full.

In addition to the planar and tomographic images captured during infusion, SPECT/CT imaging was performed at 20% of the planned therapeutic dose, end of infusion (EOI), and 24 h, 120 h, and 192 h post-treatment. Planar whole-body imaging was performed at EOI and 24 h, 72 h, 120 h, and 192 h post-treatment. Urinary excretion of radioactivity was evaluated by collecting samples of voided urine during the 48 h post-treatment. Radioactivity was measured in duplicate, with sample count rates corrected for decay and expressed as a percentage of administered activity. Blood radioactivity was evaluated by collecting samples at 0.5 h, 1 h, 2 h, 4 h, 8 h, 24 h, 48 h, 72 h, and 120 h post-treatment. In general, following infusion, patients were moved to a designated area following the hospital's radiation safety guidelines where they remained for 1–2 days for both safety monitoring and ease of sample collection and imaging; patients did not require shielding.

Image analysis was performed using MIM software (MIM Software, Inc., Cleveland, OH). The baseline high resolution 3D volumetric post-Gd T1-weighted MR image (<1.5 mm voxel dimension at each direction) was used as the primary image for tumor definition and for co-registration of SPECT/CT images[38]. In brief, gross tumor volume (GTV) was contoured on the primary baseline T1-weighted MR image. Prior to image analysis, SPECT images from each time point were co-registered to the primary MR image followed by re-saving the co-registered SPECT images to the same voxel size as the primary MR image. This provides additional detail and enhancement to allow image analysis under high resolution MRI.

To determine the volume of [186]RNL delivered to the tumor, a fixed thresholding method was used relative to maximum voxel radioactivity (count) of each SPECT image[39,40] to determine the region of interest (ROI) for the treated volume. This method considers the practical use of SPECT images to confirm that the entire tumor volume has been treated, accounting for the limited resolution of SPECT images[39]. The MIM software provided the total count in each ROI using the statistics tool and the volume of tumor in the defined treatment volume using the contour operation tool. The percentage of TVT of the SPECT image at each time point was calculated (%TuV/GTV*100).

To determine the radiation absorbed dose of [186]RNL to the tumor, radioactivity quantification of the whole brain was calculated using the total counts referenced to a standard radioactivity source[41]. Attenuation correction was performed using the CT images, validated by the total injected activity at EOI obtained from the whole-body planar gamma camera image. The radioactivity in the tumor was then determined using the percent of total tumor counts in the ROI relative to the total counts of whole brain obtained from the SPECT image at the time point. Finally, cumulated radioactivity ($\tilde{A}$) in the tumor for the entire

192 h from EOI was calculated and used to determine the mean radiation absorbed dose in the tumor volume. Whole body normal organ dose was calculated using MIRD algorithm-based OLINDA software (Hermes, Stockholm, Sweden)[41].

The radiation absorbed dose calculation uses the MIRDose algorithm, which assumes 100% of the beta-radiation dose has deposited locoregionally[41,42], while its gamma radiation dose has been neglected ([186]Re's gamma exposure is significantly lower than that of the diagnostic radionuclide, [99m]Tc[41]). This algorithm accepts that the mean path range of the therapeutic radiation particles from [186]Re is only 1.8 mm. After first calculating the cumulated radioactivity from EOI to 192 h, the radiation absorbed dose can then be determined using the following equation[41–44]:

$$D = 7.126 \, X \, \dot{A}/m$$

where $D$ is radiation absorbed dose in Gy; 7.126 is a constant in unit of Gy.g/(mCi.hr); $\dot{A}$ is cumulated radioactivity in unit of mCi.hr; and $m$ is mass of tumor in unit of g. To calculate tumor mass from volume, the density of 1 g/ml has been used. The counts of radioactivity in each normal organ are measured through drawing the ROI around each organ on anterior and posterior planar images. The count (I) in each organ is averaged from anterior and posterior images using the geometric mean method: I = SQRT(I1 × I2), where I1 is the count of an organ from the anterior planar image and I2 is the count of the organ from the posterior planar image. The total uptake of normal organs and tumors at different time intervals were computed and reported as percent of injected doses (%ID). The total body [186]Re radioactivity was computed with the summation of total body radioactivity.

CED catheters were removed ~24–48 h at the direction, timing, and discretion of the study team. This was a function of recommended timing of catheter use[15,45,46] and optimization of overall procedure time for patients. When deemed stable by the PI, patients were discharged from the hospital with precautions per the hospital's radiation and patient safety guidelines. In the absence of site-specific radiation discharge criteria, suggested study discharge criteria was as follows: At the time of discharge, the dose at 1 meter from the patient will be less than 5 mR/hr, as per standard institutional discharge criteria for radioactivity administration (for example, in thyroid cancer patients treated with [131]I). An example of patient discharge instructions in the absence of hospital-specific documentation was provided to each site.

Post-treatment evaluations to assess patient tolerability, AEs, progression, and OS were performed. Tumors were examined by MRI at baseline, week 4, and every 8 weeks thereafter. Tumor responses were evaluated by the investigators according to RANO. Volumetric assessment of tumor volume was performed using IBNeuro software (Imaging Biometrics, Elm Grove, WI). Treatment volume was assessed using MIM SurePlan MRT software (Cleveland, OH, USA).

## Trial oversight

The study was performed in accordance with ethical principles that have their origin in the Declaration of Helsinki and are consistent with International Council for Harmonisation (ICH)/GCP and applicable regulatory requirements, and was conducted in accordance with applicable national, state, and local laws. Written informed consent was obtained from each subject prior to the subject entering the study or the performance of any study related procedures. During screening, candidates received a copy of the Informed Consent Form (ICF) that was approved by the Investigator's Investigational Review Board. Each patient was fully informed about the full nature of the study, possible benefits risks, and asked for permission to use protected health information (in accordance with the Health Insurance Portability and Accountability Act or HIPAA). Candidates read and signed the ICF in the presence of a member of the study team after all patient or family

questions were answered. Refusal to sign informed consent and permission excluded an individual from the study.

Prior to advancement to the next cohort, the Data and Safety Monitoring Board (DSMB) was consulted. The DSMB is an independent group of experts that advises the PI and the study investigators. The members of the DSMB serve in an individual capacity and provide their expertise and recommendations. The primary responsibilities of the DSMB are to (1) periodically review and evaluate the accumulated study data for participant safety, study conduct, and progress, and, when appropriate, efficacy, and (2) make recommendations to the PI concerning the continuation, modification, or termination of the trial. The DSMB considers study-specific data as well as relevant background knowledge about the disease, test agent, or patient population under study. The DSMB is responsible for defining its deliberative processes, including event triggers that would call for an unscheduled review, stopping guidelines, and voting procedures prior to initiating any data review. The DSMB reviewed cumulative study data to evaluate safety, study conduct, and scientific validity and integrity of the trial. As part of this responsibility, DSMB members must be satisfied that the timeliness, completeness, and accuracy of the data submitted to them for review are sufficient for evaluation of the safety and welfare of study participants. The DSMB also assessed the performance of overall study operations and any other relevant issues, as necessary. At the conclusion of a DSMB meeting, the DSMB discussed its findings and recommendations with PI and the study investigators. The DSMB issued a written summary report that identified topics discussed by the DSMB and described their individual findings, overall safety assessment, and recommendations regarding proceeding to the next cohort as applicable.

### [186]RNL manufacturing

[186]RNL was labeled under sterile conditions at Alamo Nuclear Pharmacy Services, Inc. (San Antonio, Texas) as previously described[23], with nanoliposomes from University of Texas Health Science Center at San Antonio (UTHSCSA, San Antonio, Texas), BMEDA from ABX Advanced Biochemical Compounds, GmbH (ABX, Radeberg, Germany), and [186]Re provided and processed by University of Missouri Research Reactor (MURR, Columbia, Missouri). Unlabeled liposomes were tested for particle size distribution, pyrogenicity, and sterility prior to the radiolabeling procedure. The mean particle diameter of the liposomes was ≤130 nm with ≤0.2 polydispersity index. Final drug product was assayed for endotoxin levels and was less than 175 EU/dose prior to use. Sterility of drug product was verified as part of delayed testing along with particle size measurements. Drug product was certified by a Board-Certified Nuclear Pharmacist prior to administration.

### Statistics

Discrete outcomes were summarized with frequencies and percentages, and continuously distributed outcomes with the mean and standard deviation (SD). Absorbed dose was dichotomized to "≥100 Gy and <100 Gy" as defined by absorbed dose≥100 and absorbed dose <100, respectively. The significance of variation in the mean and cohort was assessed with analysis of variance. The significance of associations between categorical outcome and cohort was assessed with chi-square and Fisher's exact tests. Variation in survival with category of absorbed dose was assessed, without covariate adjustment, with log rank tests and with accelerated failure time (AFT) models with adjustment for covariates; a lognormal error term was assumed. All measurements were made on individual patients. No repeated measures within patient are shown. Covariates were used in the AFT models and are specified in the summary tables. Except for the AFT models, where the logarithm of survival (PFS and OS) is modeled in terms of covariates, all statistical analyses were done in original units. Measures of central tendency such as the mean and median are specified in the tables. Fold change was defined as the antilog of the

beta coefficient in the AFT model. No Bayesian methods and no hierarchical models were used. The sample sizes were specified in accordance with 3 + 3 methodology; power calculations were not made and were not used. Corrections for multiple comparisons were not applied. All significance testing was 2-sided with a significance level of 5%. SAS Version 9.4 (SAS Institute, Cary NC) was used.

### Safety

For this study, a dose limiting toxicity (DLT) was defined as grade 3 or greater acute CNS toxicity attributable to the study intervention which persists for 96 h or more (see below discussion of delayed events) OR grade 3 or greater non-CNS toxicity which is attributable to the study intervention, as per the grading scale of the grading scale of the CTCAE v4.0[35]. The standard DLT window was 28 days following treatment. Additionally, given the possibility for radiation effects outside of the standard 28-day DLT window, additional consideration was given that extended the DLT evaluation period for CNS toxicity to 90 days between successive cohorts. Lastly, if a patient within a cohort experienced a CNS toxicity that would be defined as dose limiting, the entire cohort would complete 90 days evaluation before the successive cohort would commence. In summarizing (Table 2), where more than one of the same AE or SAE has been recorded in a single patient, we eliminated multiple occurrences by using the AE or SAE with the highest grade. Percentages are computed on a per patient basis.

### Reporting summary

Further information on research design is available in the Nature Portfolio Reporting Summary linked to this article.

## Data availability

Source data are provided with this paper. All individual participant data that underlie the results reported, after de-identification, will be shared by the lead contact, A.J.B., MD, PhD (BrennerA@uthscsa.edu), upon request for at least 1 year following publication of this manuscript. The study protocol is available in the Supplementary Information file. All remaining data can be found in the Article, Supplementary and Source Data files. Source data are provided with this paper.

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

## Acknowledgements

The study was supported by an NCI Research Project Grant R01CA235800 (A.J.B.), CPRIT Commercialization Grant DP150021 (A.J.B.), as well as National Cancer Institute P30 award CA054174 (A.J.B.). We thank the site study coordinators, nuclear medicine and radiation safety staff, Neuro-surgery and Neuro-oncology physicians, Data Safety Monitoring Board, and Plus Therapeutics team who have contributed to this trial.

## Author contributions

A.J.B., M.M., A.B., J.M., N.L.F., and M.H. wrote the original manuscript. A.J.B., A.B., W.P., and J.F. conceived and designed the study. A.J.B. is the study PI, developed the study protocol, and performed all case plan-ning. For the patients included in this analysis, A.J.B., T.P., and J.S.W.

served as PIs for their respective sites. M.Y. and C.K.M. were sub-investigators who supported clinical research activities. J.M. did the statistical analysis. A.B. did the dosimetry analysis. All authors reviewed the manuscript, provided revisions, and approved the decision to submit for publication.

## Competing interests

All authors have reviewed the data analyses, contributed to data interpretation, contributed to the intellectual content of the manuscript, approved the final version to be published, and agree to be accountable for all aspects of the work. A patent for radiolabeled liposomes and their uses (US-20220273832) is assigned to Plus Therapeutics, Inc. and NanoTx, Corp., with M.H., A.J.B., A.B., and W.P. as inventors. A.B., W.P., and J.M., in addition to M.M., N.L.F., and M.H., are consultants and employees of Plus Therapeutics, Inc., respectively. Clinical trial investigation was conducted by A.J.B., T.P., M.Y., and J.S.W. Funding for the clinical trial was provided by the NIH/NCI, CPRIT, and Plus Therapeutics, Inc.
