## [Transparent Peer Review file · Nature Communications]

Treatment of Recurrent Glioma by Rhenium (^{186}Re) Obisbameda (^{186}RNL): a phase 1 clinical trial

Corresponding Author: Dr Andrew Brenner

Version 0:

Reviewer comments:

Reviewer #1

(Remarks to the Author)

This is an initial report of a dose finding study using a novel radio-ligand delivered by convection enhanced delivery for recurrent glioma. The data suggest encouraging toxicity profile and confirm the feasibility of this approach and as such are valuable. In terms of preliminary data on outcome, although this also appears promising, any conclusion needs to be heavily caveated due to patient selection and clinical and tumour variable which cannot be accounted for. This should be more robustly discussed as it lessens the impact of the manuscript. There are also some important details missing (major ones listed below) and some aspects that need more clarification.

Introduction: the standard of care referred to here is US standard, this should be caveated re other countries for first and second line treatment if the intended audience is more global.

There are no references to support the statement about technological advances in CED. These should be added.

The reference for the important pre-clinical study using beagle dogs is also not included.

In methods a succinct description of patient selection criteria is an important omission— it is not clear whether these patients were selected as they were all suitable for resection and/or CED, or based on tumour volume. This is a critical issue as the comparison used re outcome data are from studies of systemic agents, with likely very different selection criteria.

I note that the 2016 WHO grading system was used for pathology/molecular pathology diagnosis. The authors need to justify this and explain potential impact this may have in relation to the changing diagnostic criteria now used for high grade gliomas.

Table 3 is confusing - it is not clear how incidence and % are calculated, if N= 21 and % are lower than incidence then presumably this refers to multiple incidence of SAE in the same patient. Please confirm.

It is not made clear in the text how catheter number was determined during different cohorts. This is crucial since catheter number, infused tumour volume and delivered dose appear to be relevant variables.

Some practical issues are also missing - for example if these patients required shielded accommodation post operatively and if so for how long.

In the results section there is no justification for dichotomising at 100Gy, these are exploratory data and should be described as such.

In discussion an expanded consideration of why this approach is more appealing than others, especially other brachytherapy techniques would improve the context.

Reviewer #2

(Remarks to the Author)

In this Phase 1 dose escalation trial, the authors utilized a novel approach of CED of Rhenium (186Re) Obisbameda (186RNL), chelated-186Re encapsulated in nanoliposomes in 21 patients with recurrent, high-grade malignant glioma. The authors found that MTD was not reached and there were limited AEs, but meaningful median OS could be achieved if a maximum absorbed dose of 100 Gy was reached. This provocative study is a potentially meaningful contribution to the neuro-oncological community. There are several where this manuscript could be improved for the clarity and impact of this work.

Introduction

1. For context, current clinical trial use of CED in glioma should be expanded upon briefly.
2. The authors summary of their previous work is strong. However please explicitly state that a GBM model was utilized in their preclinical work if true.
3. What is the extent of the literature on local delivery of radionucleotides in glioma?
4. At its first introduction, please comment on the tumoricidal potential of rhenium-186

Results

1. It would be helpful to include an infographic summary diagram. This would include a diagram describing the nanoliposome treatment/creation, the CED strategy, outcomes assessed, etc.
2. The authors should look to improve the characterization of included patients/tumors which currently is limited.
 - a. Minimal description of previous treatments patients received should be described in the text.
 - b. Additional information in a table may also include any histological characterization on pre-treatment biopsies prior to catheter placement. Images and labeling indices for markers such as Ki67 would be helpful to better understand included patients. Images can go into supplementary material.
 - c. Where were the tumors most commonly located? Did the authors treat deeply seated, eloquent tumors?
 - d. Were these all first recurrence patients?
 - e. What selection criteria were used to limit tumor size?
3. The authors report that the number of catheters decided upon was at the discretion of the treating neurosurgeon. However, further description here would be helpful. How many catheters generally (range) were included per patient and per cohort. Were the catheters always placed inside the contrast enhancing portion of the tumor, or was the non-enhancing margin also considered?
4. Did the length of treatment vary between patients?
5. The authors report the use of whole-body planar and SPECT imaging throughout the treatment + post-treatment time points. A panel of figures would be helpful to better understand the distribution of treatment throughout the brain and in the tumor over time (during and after treatment). This could go into the main text. Did these values decrease significantly over time?
6. It is important to note the authors found that the number of catheters correlated with Vd and volume of RNL infused. What is also interesting is that it did not correlate significantly with tumor coverage. Does this mean that more catheters, does not lead to more tumor coverage? Figures summarizing these 3 findings (perhaps as supplementary figure) and discussing this in the discussion section would be of great interest to readers.
7. Were recurrences commonly found within what the authors defined as the treated tumor volume, or outside?
8. Please provide more quantitative detail about backflow.
9. Infusion rates as high as 20 ul/min were used. Was there backflow at these rates?
10. How did ECOG change pre-post treatment? The concerns of cognitive side effects of RT is well known, and local delivery may limit treatment of so-called "normal brain".
11. Supplementary Figure S1a-b is very interesting. Does this mean that a certain threshold is reached at treated volumes $\geq 70\%$ where then and only then there is a significant increase in absorbed dose? How could the authors explain this?

Discussion

1. A limitations section should be included
2. As discussed in comment above, the authors report that the number of catheters significantly correlated with Vd. This should be discussed further, given that undertreating the tumors can be a major limitation in glioma trials, including those using CED with a single catheter. Will the authors next trial aim to use multiple catheters, regardless of tumor size defined by contrast enhancement?
3. The authors correctly discuss the need to target for CE and NE tissue. Please discuss the strategy for catheter placement for future trials.
4. Pseudoprogression is a critical point the authors bring up. Post-treatment biopsies on catheter removal could be critical to understand treatment response in this context, and are a major advantage to using CED given the direct access to tissue.
5. Was the 100 Gy absorbed efficacy dose threshold all from cohort 6? Please clarify.

Methods

1. There needs to be more information about the CED treatment.
 - a. Clearly state treatment duration as it is unclear in the current form
 - b. Describe the surgical method of catheter placement
 - c. How was backflow monitored?
2. Why were catheters placed and removed on average 1-2 days before treatment started or ended?
3. Please briefly expand upon the in silico treatment planning and convection simulation.

Reviewer #3

(Remarks to the Author)

Nature Communications – Reviewing

Convection Enhanced Delivery of Rhenium (^{186}Re) Obisbameda (^{186}RNL) in Recurrent Glioma: a multicenter, single arm, phase 1 clinical trial

The authors evaluated the safety, pharmacokinetics, and preliminary efficacy of a new radiotherapeutic product, Rhenium (^{186}Re) Obisbameda (^{186}RNL), in a multicenter, single arm, phase 1 study for the treatment of recurrent glioma. The authors have developed their strategy for re-irradiation in recurrent WHO Grade 3 or 4 glioma using ^{186}Re -nanoliposomes based on pre-clinical data. By directly targeting tumors using convection enhanced delivery (CED), they were able to safely deliver 22.3 mCi in 8.80 ml using up to 4 catheters. The doses of radiation delivered are much higher than those obtainable with conventional external radiotherapy. In this initial analysis, the maximum tolerated dose (MTD) was not reached nor was the dose-limiting toxicity (DLT) and overall survival (OS) was shown to be closely correlated with tumor percent volume coverage by the drug and the dose of radiation absorbed by the tumor. This is a novel approach in an unmet need cancer. It is very appealing to use a tracer that can also be used to image agent distribution in brain. I have several comments and suggestions for revision:

- 1) Although the method is entirely sound from a neurosurgical point of view and the link with current standards of care is relevant, it would be useful to clarify where this type of treatment can be developed and how it can or cannot be extended everywhere. Also, the cost of such support must be raised through a study of the corresponding economic mode; at least be mentioned.
- 2) 21 patients are included in the study, divided into two groups according to the dose deposited (>100 Gy $n=12$; <100 Gy $n=9$), demonstrating a survival advantage in patients who received the highest dose: 17 months versus 6 months. These figures are compared to the 8 months obtained for the standard of care. In general, on this subject, it is 8 months but without progression, which is not the case here since it involves recurrences. Can the authors clarify this?
- 3) The choice of ^{186}Re (with a half-life of ~ 89.2 hours) is made here. It is a reactor-produced isotope that can also be generated by cyclotron. Can the authors comment on the interest, advantages and disadvantages of this product. On this subject, the constraint for clinicians and patients (37 days for zero radioactivity) is not mentioned? Could the same type of work have been done with ^{188}Re which is a generator product and has a half-life of ~ 16.9 hours (total decay in 7 days)?
- 4) In link with the above, it is essential to discuss the choice of the radiopharmaceutical (^{186}Re -nanoliposome). In this regard, it is important to highlight recent preclinical developments more broadly on the keywords: radiotherapy, rhenium, nanoparticle, glioma (eg. DOI: 10.7150/thno.19403; DOI: 10.1016/j.biomaterials.2011.05.067).
- 5) The radiation absorbed dose calculation uses the MIRDose algorithm which assumes 100% of the beta-radiation dose has deposited locoregionally. It is not perfectly clear how the dose distribution study attest to this point (volume of ^{186}RNL distributed in the brain and in the tumor, percentage of tumor volume treated, radiation absorbed dose to the tumor, and whole-body normal organ doses)?
- 6) Regarding, MGMT status, it has indeed been demonstrated that it impacts the effectiveness of temozolomide (TMZ), which however was not used in this study. Does the author have any evidence that it can define ^{186}Re -radioresponse which will give echoes of how the data produced here on this subject could be further used (larger cohorts) in terms of prognosis feature.
- 7) Although this reviewer is not a statistician, it seems important to include the contribution of an expert in the field in this study. For example, there is no mention of calculated sample size, hazard ratio (HR) or randomization. Can you clarify this point ?

Reviewer #4

(Remarks to the Author)

The authors reported the results of the Phase I dose escalation trial NCT01906385 that evaluated a single dose of ^{186}RNL administered through a convection enhanced delivery catheter in participants with recurrent Glioma (GBM). The study employed a modified 3+3 Fibonacci dose escalation method. The maximum tolerated dose (MTD) was not reached, and overall survival was found to correlate with both the percentage of tumor volume coverage by the drug and the radiation absorbed dose to the tumor.

The manuscript is well-written. We suggest adding a few more details on the study design and results for rigor and reproducibility, and for interpretation of the results, as recommended by the Dose Finding CONSORT extension (DEFINE-CONSORT; BMJ 2023 Oct 20;383:e076387. doi: 10.1136/bmj-2023-076387.).

The definitions of the endpoints should be included and/or clarified:

- o The DLT definition, even if it follows the standard definition based on grade 3 of the CTCAE scale, should be clearly stated, including the DLT evaluation window.
- o The time point for the absorbed dose could be stated more clearly.
- o The timepoint at which the volume of distribution was obtained can also be clarified.

- While the paper mentions that the 3+3 design is used, it does not clearly state the dose-escalation decisions and the definition of MTD. For example,
 - o Whether other endpoints were considered for the dose escalation (e.g., pharmacokinetic measures).
 - o The 3+3 would not have 6 patients in cohort 6 in the absence of DLT. What was the rationale for adding three more patients in cohort 6 (with increasing maximum flow rate)? Was this pre-specified in the protocol?
- The dose levels to be evaluated should be specified in the methods along with the rationale for them and the starting dose. It is not clear if:
 - o The Fibonacci was used on RNL activity and the infused volume was calculated from that
 - o The methods should also clarify the relationship between the assigned and absorbed dose.
- Whether a sentinel patient was included or not.

We appreciated the results presented both by cohort and by the total number of trial patients. In general, the results are well-reported and discussed. Further clarifications on:

- The number of catheters used by cohort and in total should be reported as this was left to the discretion of the treating physician and it is not clear if it differed by cohort.
- Report the AE by cohort using patient as unit of analysis to know what proportion of patients experienced those grades of AE in each cohort. The frequency of events using events are unit (Tables 4 and 5) can be supplemental.
- Summary statistics on the absorbed doses per cohort.
- In Figure 1 (page 6), while the median Vd in cohort 6 is higher compared to all other cohorts, the maximum Vd is observed in a patient in cohort 5. Do the authors have an explanation for this?
- For survival analysis, was the dichotomization (<100 Gy or not) decided before the study onset, or is it a post-hoc analysis? Why was the survival analysis not done by assigned dose/cohort like ORR, but by absorbed dose?
- For Table S4, did the authors examine the correlation between “Ratio of Treated to Total Tumor Volume”, “Total Dose in Distribution Volume”, and “Volume Administered” for the “Progression-Free Survival Accelerated Failure Time Model”? It is not clear why both of this need to be in the model.
- The trial accrued more males than females and mostly white patients; this could be discussed in terms of generalizability.

Reviewer #5

(Remarks to the Author)

Version 1:

Reviewer comments:

Reviewer #1

(Remarks to the Author)

The authors have addressed the review comments very thoroughly. This has added much to enhance the clinical relevance of the work and to set it in better context.

Reviewer #2

(Remarks to the Author)

The authors have satisfactorily addressed the comments.

Reviewer #3

(Remarks to the Author)

Although the authors have generally done their best to improve the manuscript, question number 4 of this reviewer was not addressed from the angle necessary to understand the interest of the radiopharmaceutical (radiolabeled nanoliposome or rhenium-loaded nanoliposome).

Major : The advantages of the vector were not presented as a possible alternative considering other colloids in the literature (biodistribution profiles, stability, safety). The four reviews cited are not directly related to the choice of the nanoliposome and its performances. They mainly consider static implants tested in the clinic but not dynamic nanovectors also in development. Why this one and not another? Would another colloid have achieved the same result? What about what is available from preclinical to clinical?

This point addressed, I recommend publication.

Reviewer #4

(Remarks to the Author)

The authors have clarified some of the previous questions. However, based on their clarifications, I have the following comments and suggestions:

1. Based on the revision, "Given the lack of significant AEs in the first 4 cohorts, the DSMB, dose doubling was recommended." The manuscript should state that cohorts 5 and 6 were added later after review of data from first 4 cohorts as doses in phase 1 trials are generally pre-specified for rigor and reproducibility. Please specify the set of doses that were specified at the start of the trial in the treatment section and those that were added after review and the rationale for those dose selection.
2. Table 2 for the AE is still by events not patients. The unit of analysis is still events to have 24 mild events in the first cohort with 3 patients. Please specify the number of AE out of the 3 patients taking the maximum grade of AE.
3. The variability in the absorbed dose is of concern (Table S8) and not appropriate to summarize at mean +/- SEM in Figure 1. Given the small sample size and large variability per cohort it is best to graph all points. Please explain the poor correlation between administered dose and absorbed dose. Was this evaluated? How reproducible is the absorbed dose?
4. It is important to note that survival was not associated with administered dose and only with absorbed dose in the paper. This is very important and should be clearly explained and discussed. Absorbed dose is an outcome which could be dependent on many factors.

Reviewer #5

(Remarks to the Author)

Reviewer #6

(Remarks to the Author)

Reviewer #7

(Remarks to the Author)

Version 2:

Reviewer comments:

Reviewer #3

(Remarks to the Author)

The manuscript deserves publication in Nature Communications considering the improvements made by the authors.

Reviewer #4

(Remarks to the Author)

While the most of the new revisions have addressed my previous comments, I think it is important to address question 4 and note in the results and discussion that survival outcomes were not associated with administered dose. This is very important and worthy to be discussed as it is relevant to the reproducibility of a dose-finding clinical trial.

Reviewer #1

We thank Reviewer #1 for their thoughtful consideration of our manuscript and provide the following supporting information to address their open questions.

1. Introduction: the standard of care referred to here is US standard, this should be caveated re other countries for first and second line treatment if the intended audience is more global.

We have revised the text in the Introduction section to address standard of care, by stating: *Standard of care (SoC) for patients with recurrent glioblastoma is not well defined⁶. Median overall survival (mOS) with first progression is typically ~7.4-9.2 months^{7,8} and with second progression, approximately four months⁹.*

The following references were used as support:

- Weller, M. et al. European Association for Neuro-Oncology (EANO) guideline on the diagnosis and treatment of adult astrocytic and oligodendroglial gliomas. *Lancet Oncol* 18, e315–e329 (2017).
- Friedman, H. S. et al. Bevacizumab Alone and in Combination With Irinotecan in Recurrent Glioblastoma. *JCO* 27, 4733–4740 (2009).
- Cloughesy, T. F. et al. A randomized controlled phase III study of VB-111 combined with bevacizumab vs bevacizumab monotherapy in patients with recurrent glioblastoma (GLOBE). *Neuro-Oncology* 22, 705–717 (2020).
- Quant, E. C. et al. Role of a second chemotherapy in recurrent malignant glioma patients who progress on bevacizumab. *Neuro-Oncology* 11, 550–555 (2009).

2. There are no references to support the statement about technological advances in CED. These should be added.

We have revised the text in the Introduction section to address technological advances in CED, by stating: *A number of clinical trials have been conducted using CED primarily with chemotherapeutics or targeted toxins. The first randomized, Phase III evaluation of an agent administered via CED was cintredekin besudotox versus Gliadel wafers in the PRECISE trial¹⁴. While negative, a retrospective analysis of the expected drug distribution based on catheter positioning data revealed that only 49.8% of catheters met all positioning criteria and overall survival ($p = 0.006$) was higher for investigators considered experienced after adjusting for patient age and Karnofsky Performance Scale score¹⁵. However, since that time a number of technological improvements have been implemented including improved catheter design, neuro-navigation, and in silico planning^{16–18}.*

The following references were used as support:

- Kunwar, S. et al. Phase III randomized trial of CED of IL13-PE38QQR vs Gliadel wafers for recurrent glioblastoma. *Neuro Oncol* 12, 871–881 (2010).
- Sampson, J. H. et al. Poor drug distribution as a possible explanation for the results of the PRECISE trial. *J Neurosurg* 113, 301–309 (2010).
- Lonser, R. R. et al. Successful and safe perfusion of the primate brainstem: in vivo magnetic resonance imaging of macromolecular distribution during infusion. *J Neurosurg* 97, 905–913 (2002).
- Asfaw, Z. K., Young, T., Brown, C. & Germano, I. M. Charting the success of neuronavigation in brain tumor surgery: from inception to adoption and evolution. *J Neurooncol* (2024) doi:10.1007/s11060-024-04778-0.
- Wembacher-Schroeder, E. et al. Evaluation of a patient-specific algorithm for predicting distribution for convection-enhanced drug delivery into the brainstem of patients with diffuse intrinsic pontine glioma. *J Neurosurg Pediatr* 28, 34–42 (2021).

3. The reference for the important pre-clinical study using beagle dogs is also not included.

The nonclinical safety and toxicity study was commissioned in advance of the IND submission and not published but reviewed by the NIH/NCI in the grant application that supported the trial. We can make this available on request as noted in the Data Availability statement. We have revised the text in the Introduction section to make note this was unpublished by stating: *A preclinical laboratory study conducted in accordance with the United States Food and Drug Administration (FDA) Good Laboratory Practice (GLP) Regulations was subsequently conducted to assess overall toxicity and evaluate dosimetry of a single dose administration of ¹⁸⁶RNL (unpublished reference data).*

4. In methods a succinct description of patient selection criteria is an important omission– it is not clear whether these patients were selected as they were all suitable for resection and/or CED, or based on tumour volume. This is a critical issue as the comparison used re outcome data are from studies of systemic agents, with likely very different selection criteria.

We have revised the text in the Eligibility section (Methods) to state: *Patients were enrolled at two study sites: UT Health San Antonio and UT Southwestern Medical Center. Eligible participants were at least 18 years of age, able to provide written consent and had histologically confirmed recurrent WHO Grade 3 or 4 glioma with an enhancing tumor volume within the treatment field volume in the respective cohort, based on a the most recent (within 35 days) magnetic resonance imaging (MRI) assessment. The 2016 WHO classification²⁸ was utilized throughout the study. Only two patients (one each from cohort 4 and 5) had the IDH1 mutation (Table 1), making them not glioblastoma under the current 2021 WHO classification², which did not meaningfully alter the findings of this study and were therefore included in the analysis. Patients were not limited by their number of recurrences. Recurrences were defined as progression by RANO criteria or other clinically accepted neuro-oncology evaluation. Patients were further limited to having completed standard treatment options with known survival benefit for any recurrence (e.g., surgery, temozolomide, radiation, and tumor treating fields), but were allowed on study if medically unable or unwilling to follow standard treatment options for any recurrence. Additionally, the protocol was modified after cohort 4 to exclude patients with Grade 3 glioma or prior bevacizumab treatment, as some of the early patients with prior bevacizumab therapy showed poor ¹⁸⁶RNL convection to the tumor, lower tumor coverage, and lower tumor absorbed dose. Progression was determined by Radiographic Assessment in Neuro-Oncology (RANO) criteria³⁰ following standard treatment.*

Likewise, we have revised the text in the Treatment section (Methods) to state: *Once patients were consented, they underwent treatment planning MRI within seven days before drug infusion to evaluate tumor characteristics, including location, structure, shape, and dimension. Based on the treatment planning MRI, patients were further selected for study status. If the tumor had progressed beyond the screening criteria, they were considered a screen failure.*

The following references were used as support:

- Louis, D. N. et al. The 2016 World Health Organization Classification of Tumors of the Central Nervous System: a summary. *Acta Neuropathol* 131, 803–820 (2016).
- Louis, D. N. et al. The 2021 WHO Classification of Tumors of the Central Nervous System: a summary. *Neuro-Oncology* 23, 1231–1251 (2021).
- Wen, P. Y. et al. Updated Response Assessment Criteria for High-Grade Gliomas: Response Assessment in Neuro-Oncology Working Group. *JCO* 28, 1963–1972 (2010).

We would also like to bring attention to a recent analysis we performed together with Medidata with aggregate summary comparing bevacizumab and CED patients based on combined study- and patient-level data using weighted (by sample size) means, incidence, and overall survival (OS) rates and median of medians. This included CED studies referenced in D’Amico (*J Neurooncol* 2021) and Medidata Enterprise Data Store (MEDS). This resulted in 163 recurrent glioblastoma bev patients identified from MEDS and a CED cohort including 636 patients from MEDS and studies referenced in D’Amico. We observed no clear difference in survival between these cohorts suggesting that patients enrolled on CED studies (such as ours) can be compared to historical studies of systemic therapy (Ensign, *J Clin Oncol* 2023).

The following references were used as support:

- D’Amico, R. S., Aghi, M. K., Vogelbaum, M. A. & Bruce, J. N. Convection-enhanced drug delivery for glioblastoma: a review. *J Neurooncol* 151, 415–427 (2021).
- Ensign, L.G., Boisvert, D., LaFrance N., Hendershot, J., Michalek, J., Brenner A.J., Davi, R. Clinical characterization of patients with recurrent glioblastoma in trials involving CED and non-CED treatment. *J Clin Oncol* 2023 41:16_suppl, e18845-e18845.

5. I note that the 2016 WHO grading system was used for pathology/molecular pathology diagnosis. The authors need to justify this and explain potential impact this may have in relation to the changing diagnostic criteria now used for high grade gliomas.

We have revised the text in the Eligibility section (Methods) to state: *The 2016 WHO classification²⁸ was utilized throughout the study. Only two patients (one each from cohort 4 and 5) had an IDH1 mutation (Table 1), making them not glioblastoma*

under the current 2021 WHO classification², which did not meaningfully alter the findings of this study and were therefore included in the analysis.

The following references were used as support:

- Louis, D. N. et al. The 2021 WHO Classification of Tumors of the Central Nervous System: a summary. *Neuro-Oncology* 23, 1231–1251 (2021).
- Louis, D. N. et al. The 2016 World Health Organization Classification of Tumors of the Central Nervous System: a summary. *Acta Neuropathol* 131, 803–820 (2016).

6. Table 3 is confusing - it is not clear how incidence and % are calculated, if N= 21 and % are lower than incidence then presumably this refers to multiple incidence of SAE in the same patient. Please confirm.

Thank you for identifying this. We have revised the text in the Safety section (Methods) to state: *In summarizing (Table 2), where more than one of the same AE or SAE has been recorded in a single patient, we eliminated multiple occurrences by using the AE or SAE with the highest grade. Percentages are computed on a per patient basis.* Likewise, we have updated the AE Table 2 and Results sections accordingly.

7. It is not made clear in the text how catheter number was determined during different cohorts. This is crucial since catheter number, infused tumour volume and delivered dose appear to be relevant variables.

We have revised the text in the Treatment section (Methods) to state: *Once patients were consented, they underwent treatment planning MRI within seven days before drug infusion to evaluate tumor characteristics, including location, structure, shape, and dimension. Based on the treatment planning MRI, patients were further selected for study status. If the tumor had progressed beyond the screening criteria, they were considered a screen failure. Once confirmed to be appropriate for CED, stereotactic treatment planning was centralized and performed by the Principal Investigator (PI) together with the local neurosurgeon. The tumor volume was calculated by the use of iPlan Flow, as was all in silico treatment planning and convection simulation, as has been previously described¹⁸. Briefly, using the iPlan software, the PI would determine the number of catheters and their trajectories, based on the size and location of the tumor, in accordance with pre-set parameters noted below. Early cohorts minimized the number of catheters for safety and to characterize the distribution; once this was determined, the number of catheters used (1-4) was based on planning to allow for suitable coverage of the enhancing tumor with surrounding T2 FLAIR abnormality.*

The following reference was used as support:

- Wembacher-Schroeder, E. et al. Evaluation of a patient-specific algorithm for predicting distribution for convection-enhanced drug delivery into the brainstem of patients with diffuse intrinsic pontine glioma. *J Neurosurg Pediatr* 28, 34–42 (2021).

8. Some practical issues are also missing - for example if these patients required shielded accommodation post operatively and if so for how long.

We have revised the text in the Treatment section (Methods) to state: *In general, following infusion, patients were moved to a designated area following the hospital's radiation safety guidelines where they remained for 1-2 days for both safety monitoring and ease of sample collection and imaging; patients did not require shielding.*

Likewise, we have included the following further down in that section: *CED catheters were removed ~24-48 hours at the direction, timing, and discretion of the study team. This was a function of recommended timing of catheter use^{15,39,40} and optimization of overall procedure time for patients. When deemed stable by the PI, patients were discharged from the hospital with precautions per the hospital's radiation and patient safety guidelines. In the absence of site-specific radiation discharge criteria, suggested study discharge criteria was as follows: At the time of discharge, the dose at 1 meter from the patient will be less than 5 mR/hr, as per standard institutional discharge criteria for radioactivity administration (for example, in thyroid cancer patients treated with ¹³¹I). An example of patient discharge instructions in the absence of hospital-specific documentation was provided to each site.*

The following references were used as support:

- Sampson, J. H. *et al.* Poor drug distribution as a possible explanation for the results of the PRECISE trial. *J Neurosurg* 113, 301–309 (2010).
- Thompson, E. M. *et al.* Recombinant polio–rhinovirus immunotherapy for recurrent paediatric high-grade glioma: a phase 1b trial. *The Lancet Child & Adolescent Health* 7, 471–478 (2023).
- Mueller, S. *et al.* PNOC015: Repeated convection-enhanced delivery of MTX110 (aqueous panobinostat) in children with newly diagnosed diffuse intrinsic pontine glioma. *Neuro-Oncology* 25, 2074–2086 (2023).

9. In the results section there is no justification for dichotomising at 100Gy, these are exploratory data and should be described as such. In discussion an expanded consideration of why this approach is more appealing than others, especially other brachytherapy techniques would improve the context.

In the Introduction section, we state: *Furthermore, >100 Gy of ¹⁸⁶RNL was shown to eradicate grafted tumors and prolong overall survival, without clinical or microscopic evidence of toxicity²⁰.*

We have also revised the Efficacy section (Results) to state: *Given the findings from our preclinical studies identifying 100Gy as a threshold for survival benefit²⁰, patients were also dichotomized by absorbed dose of 100 Gy for PFS and OS.*

The following reference was used as support:

- Phillips, W. T. *et al.* Rhenium-186 liposomes as convection-enhanced nanoparticle brachytherapy for treatment of glioblastoma. *Neuro-Oncology* 14, 416–425 (2012).

We have also revised the text in the Statistics section to better describe the dichotomization, stating: *Absorbed dose was dichotomized to “ ≥ 100 Gy and < 100 Gy” as defined by $\text{absorbed dose} \geq 100$ and $\text{absorbed dose} < 100$, respectively.*

We have also revised the text in the Discussion section to state: *Ideal isotopes in CNS brachytherapy, as demonstrated herein with ¹⁸⁶Re, should have several features including short path length, low dose rate, high energy output, and a means to image and compute dose. Furthermore, the isotope should address both enhancing and non-enhancing tumor and remain in place throughout the decay cycle of the isotope to minimize systemic exposure.*

Reviewer #2

We thank Reviewer #2 for their thoughtful consideration of our manuscript and provide the following supporting information to address their open questions.

Introduction

1. For context, current clinical trial use of CED in glioma should be expanded upon briefly.

We have revised the text in the Introduction section to address technological advances in CED, by stating: *A number of clinical trials have been conducted using CED primarily with chemotherapeutics or targeted toxins. The first randomized, Phase III evaluation of an agent administered via CED was cintredekin besudotox versus Gliadel wafers in the PRECISE trial¹⁴. While negative, a retrospective analysis of the expected drug distribution based on catheter positioning data revealed that only 49.8% of catheters met all positioning criteria and overall survival ($p = 0.006$) was higher for investigators considered experienced after adjusting for patient age and Karnofsky Performance Scale score¹⁵. However, since that time a number of technological improvements have been implemented including improved catheter design, neuro-navigation, and in silico planning^{16–18}.*

The following references were used as support:

- Kunwar, S. *et al.* Phase III randomized trial of CED of IL13-PE38QQR vs Gliadel wafers for recurrent glioblastoma. *Neuro Oncol* 12, 871–881 (2010).
- Sampson, J. H. *et al.* Poor drug distribution as a possible explanation for the results of the PRECISE trial. *J Neurosurg* 113, 301–309 (2010).
- Lonser, R. R. *et al.* Successful and safe perfusion of the primate brainstem: in vivo magnetic resonance imaging of macromolecular distribution during infusion. *J Neurosurg* 97, 905–913 (2002).

- Asfaw, Z. K., Young, T., Brown, C. & Germano, I. M. Charting the success of neuronavigation in brain tumor surgery: from inception to adoption and evolution. *J Neurooncol* (2024) doi:10.1007/s11060-024-04778-0.
- Wembacher-Schroeder, E. *et al.* Evaluation of a patient-specific algorithm for predicting distribution for convection-enhanced drug delivery into the brainstem of patients with diffuse intrinsic pontine glioma. *J Neurosurg Pediatr* 28, 34–42 (2021).

2. The authors summary of their previous work is strong. However please explicitly state that a GBM model was utilized in their preclinical work if true.

We have revised the text in the Introduction to state: *In orthotopic xenograft models of glioblastoma we observed high in vivo stability characterized by slow blood clearance and gradually increasing spleen accumulation²⁰. Furthermore, >100 Gy of ¹⁸⁶RNL was shown to eradicate grafted tumors and prolong overall survival, without clinical or microscopic evidence of toxicity²⁰.*

The following reference was used as support:

- Phillips, W. T. *et al.* Rhenium-186 liposomes as convection-enhanced nanoparticle brachytherapy for treatment of glioblastoma. *Neuro-Oncology* 14, 416–425 (2012).

3. What is the extent of the literature on local delivery of radionucleotides in glioma?

We have revised the Introduction section to incorporate the key publications that informed the authors prior to study start, stating: *Prior to study start we searched PubMed from any time up to December 1, 2013 for clinical therapeutic studies using ¹⁸⁶Re to treat glioma using the search terms “(¹⁸⁶Re[ALL FIELDS] AND glioma[ALL FIELDS]) and (rhenium-186[ALL FIELDS] AND glioma[ALL FIELDS]). The search gave four publications^{23–26}. Intramedullary cystic spinal cord pilocytic astrocytoma was managed with minor side effects using ¹⁸⁶Re intracavitary irradiation, with stabilization of the cyst and neurological deficit improvement²³. Combination treatment for pilocytic astrocytoma using ¹⁸⁶Re delivered 400 Gy to the cyst wall and resulted in progressive cyst disappearance and mural nodule retraction²⁶. A fibrin glue of ¹⁸⁸Re and ¹⁸⁶Re bound in microspheres was used post-tumor resection in a 9L-glioblastoma rat model. 60% of treated animals survived 36 days, compared to control animals (17 ± 3 days)²⁴. Lastly, intracavitary ¹⁸⁶Re application was used in six cases of cystic craniopharyngiomas but abandoned because of cyst recurrence and leakage²⁵.*

The following references were used as support:

- Colnat-Coulbois, S., Klein, O., Braun, M., Thouvenot, P. & Marchal, J.-C. Management of Intramedullary Cystic Pilocytic Astrocytoma With Rhenium-186 Intracavitary Irradiation: Case Report. *Neurosurgery* 66, E1023–E1024 (2010).
- Häfeli, U. O., Pauer, G. J., Unnithan, J. & Prayson, R. A. Fibrin glue system for adjuvant brachytherapy of brain tumors with ¹⁸⁸Re and ¹⁸⁶Re-labeled microspheres. *Eur J Pharm Biopharm* 65, 282–288 (2007).
- Gahbauer, H. *et al.* Combined use of stereotaxic CT and angiography for brain biopsies and stereotaxic irradiation. *AJNR Am J Neuroradiol* 4, 715–718 (1983).
- Proust, F. *et al.* [Combination treatment for pilocytic astrocytoma: stereotaxic radiosurgery and endocavitary radiotherapy]. *Neurochirurgie* 44, 50–54 (1998).

4. At its first introduction, please comment on the tumoricidal potential of rhenium-186.

We have revised the text in the Introduction section to state: *¹⁸⁶Re is a beta emitter which exerts dose dependent therapeutic effects by inducing single strand DNA breaks²¹.*

The following reference was used as support:

- Khazaei Monfared, Y. *et al.* DNA Damage by Radiopharmaceuticals and Mechanisms of Cellular Repair. *Pharmaceutics* 15, 2761 (2023).

Results

1. It would be helpful to include an infographic summary diagram. This would include a diagram describing the nanoliposome treatment/creation, the CED strategy, outcomes assessed, etc.

We have added an infographic (Figure S4).

2. The authors should look to improve the characterization of included patients/tumors which currently is limited.

- Minimal description of previous treatments patients received should be described in the text.

We have revised the text in the Results section to state: *For prior treatments reported, 20 (95.2%) received temozolomide, 19 (90.5%) received radiation, 20 (95.2%) received surgery, and 6 (28.6%) received prior bevacizumab.*

- Additional information in a table may also include any histological characterization on pre-treatment biopsies prior to catheter placement. Images and labeling indices for markers such as Ki67 would be helpful to better understand included patients. Images can go into supplementary material.

We appreciate Reviewer 2's suggestion, however, we did not collect the pretreatment or posttreatment biopsies centrally for further analysis but will plan to do so in the future to better understand our patient population. The total information collected in the study on tumor characterization is provided in the manuscript/supplementary information.

- Where were the tumors most commonly located? Did the authors treat deeply seated, eloquent tumors?

We have revised the text in the Demographics section (Results) to state: *Tumor locations were frontal in 5 (24%), parietal in 3 (14%), temporal in 6 (29%), occipital in 2 (9%), and overlapping lobes in 5 (24%).* Infratentorial tumors were excluded at the request of the FDA clinical reviewers at the time of the IND review.

- Were these all first recurrence patients?

We have revised the text in the Demographics section (Results) to state: *13 (62%) patients were first recurrence, 7 (33%) second recurrence, and 1 (5%) third recurrence or later.*

- What selection criteria were used to limit tumor size?

We have revised the text in the Eligibility section (Methods) to state: *Eligible participants were at least 18 years of age, able to provide written consent and had histologically confirmed recurrent WHO Grade 3 or 4 glioma with an enhancing tumor volume within the treatment field volume in the respective cohort, based on a the most recent (within 35 days) magnetic resonance imaging (MRI) assessment.*

3. The authors report that the number of catheters decided upon was at the discretion of the treating neurosurgeon. However, further description here would be helpful. How many catheters generally (range) were included per patient and per cohort. Were the catheters always placed inside the contrast enhancing portion of the tumor, or was the non-enhancing margin also considered?

We revised the text in the Treatment section (Methods) to state: *Once patients were consented, they underwent treatment planning MRI within seven days before drug infusion to evaluate tumor characteristics, including location, structure, shape, and dimension. Based on the treatment planning MRI, patients were further selected for study status. If the tumor had progressed beyond the screening criteria, they were considered a screen failure. Once confirmed to be appropriate for CED, stereotactic treatment planning was centralized and performed by the Principal Investigator (PI) together with the local neurosurgeon. The tumor volume was calculated by the use of iPlan Flow, as was all in silico treatment planning and convection simulation, as has been previously described¹⁸. Briefly, using the iPlan software, the PI would determine the number of catheters and their trajectories, based on the size and location of the tumor, in accordance with pre-set parameters noted below. Early cohorts minimized the number of catheters for safety; once this was determined, the number of catheters used (1-4) was at the treating investigator's discretion based on planning to allow for suitable coverage of the enhancing tumor with surrounding T2 FLAIR abnormality. Immediately following a SoC stereotactic biopsy procedure for confirmation of disease progression, BrainLab Flexible CED catheter(s) (7.5 mm tip) were placed, with at least one catheter*

proceeding along the same needle track as the SoC biopsy. Catheter placement was optimized to avoid ependymal surfaces by >0.5 cm as they offer no resistance to fluid flow, fissure, resection cavity, or cortical surface. Distal placement of the infusing tip within the tumor was also critical to avoid the pressure gradient from the tumor core to periphery to adjacent brain. Postoperative head CT was performed to evaluate for hematoma or pneumocephalus.

The following reference was used as support:

- Wembacher-Schroeder, E. *et al.* Evaluation of a patient-specific algorithm for predicting distribution for convection-enhanced drug delivery into the brainstem of patients with diffuse intrinsic pontine glioma. *J Neurosurg Pediatr* 28, 34–42 (2021).

4. Did the length of treatment vary between patients?

We revised the text in the Treatment section (Methods) to state: *The total time of infusion was a function of flow rate, number of catheters used, and total volume infused (Table S7).*

5. The authors report the use of whole-body planar and SPECT imaging throughout the treatment + post-treatment time points. A panel of figures would be helpful to better understand the distribution of treatment throughout the brain and in the tumor over time (during and after treatment). This could go into the main text. Did these values decrease significantly over time?

We appreciate the reviewer's suggestion, and we agree that the imaging should be included in the manuscript main text. We consolidated the tables allowing us to add two figures, one with SPECT/CT of the distribution and retention at 24hrs and 5 days (new Figure 2), and a whole body planar image showing the retention out to several days and lack of activity in the remainder of the body (new Figure 3). We did not observe any redistribution after 24hrs with activity being decreased related to decay rather than change in distribution.

6. It is important to note the authors found that the number of catheters correlated with Vd and volume of RNL infused. What is also interesting is that it did not correlate significantly with tumor coverage. Does this mean that more catheters, does not lead to more tumor coverage? Figures summarizing these 3 findings (perhaps as supplementary figure) and discussing this in the discussion section would be of great interest to readers.

We revised the text in the Discussion section to state: *Although we found a correlation between the number of catheters, volume of distribution, and the administered volume, we did not observe a correlation between number of catheters and percent tumor coverage. There are a number of confounders that may impact correlation between number of catheters and percent tumor coverage, particularly that treatment planning for larger tumors likely would drive a decision to place a higher number of catheters, but that the inherent challenges of larger tumors would make coverage less successful. Coverage of greater than 70% appeared to be associated with benefit (Figure S1b), but this is likely a factor of the fixed thresholding method used relative to maximum voxel radioactivity (count) of each SPECT image, and that treatment effects extend beyond this region which may be better discriminated with 3D dosimetry.*

7. Were recurrences commonly found within what the authors defined as the treated tumor volume, or outside?

We revised the text in the Discussion section to state: *Finally, determining response can be challenging in CNS tumors, particularly with radiation which can result in pseudo-progression. To better characterize response, we are performing advanced imaging modalities including Delta T1 subtraction mapping, DSC perfusion, and delayed contrast mapping (TRAMs). In preliminary parametric analyses we observed a statistically significant difference in repeated pair measurements with untreated tumor volume significantly increased relative to treated tumor volume ($p < 0.0001$) suggesting that progression arises outside the treated volume.*

8. Please provide more quantitative detail about backflow.

We have revised the text in the Treatment section (Methods) to state: *Dynamic images were acquired via real-time persistence scope for evaluation of focal accumulation of activity at the assumed tip of the catheter(s). When activity was observed to accumulate focally, that time was designated as the beginning of the planned therapeutic volume infusion to correct for dead space in the catheter line. Catheters deemed unsatisfactory (backflow along the catheter or spillage into adjacent CSF space) were stopped and the remaining volume switched to the remaining catheters. To minimize backflow*

concerns³¹, we utilized the Brainlab catheter, which is both flexible and has a “step design” at the tip, avoided resection cavities and pial surfaces, and employed a slow ramping of infusion rates (Table S6). Furthermore, backflow was monitored during infusion using planar imaging as mentioned above, allowing changes to be made in real-time to ensure the planned administered dose was delivered in full. Additionally, we include in the results section the following: Three patients had catheter “failure,” where it was observed that one catheter did not have the same flow as the others due to a blockage (“kink”) in the delivery tubing or drug was being lost to an area of lower resistance; in these cases, all patients received the full amount of administered dose by diverting the volume of the failed catheter to another catheter in use. Therefore backflow was not a common issue in this study.

The following reference was used as support:

- D’Amico, R. S., Aghi, M. K., Vogelbaum, M. A. & Bruce, J. N. Convection-enhanced drug delivery for glioblastoma: a review. *J Neurooncol* 151, 415–427 (2021).

9. Infusion rates as high as 20 uL/min were used. Was there backflow at these rates?

Please see our response to Question 8 (“Results”). Backflow was not a common issue, was not rate dependent, but rather from inadequate catheter depth where investigators attempted to infuse the more superficial portion of the tumor. Since dynamic imaging allows live visualization of the convection progress, when this occurred drug could be redistributed to the other catheters.

10. How did ECOG change pre-post treatment? The concerns of cognitive side effects of RT is well known, and local delivery may limit treatment of so-called “normal brain”.

Patients were assessed neurologically at each visit and data was captured as adverse events as noted. It is very difficult to discriminate changes between disease and treatment effects; as such we did not report ECOG status before and after as a standalone data point.

11. Supplementary Figure S1a-b is very interesting. Does this mean that a certain threshold is reached at treated volumes $\geq 70\%$ where then and only then there is a significant increase in absorbed dose? How could the authors explain this?

We revised the text in the Discussion section to state: *Coverage of greater than 70% appeared to be associated with benefit (Figure S1b), but this is likely a factor of the fixed thresholding method used relative to maximum voxel radioactivity (count) of each SPECT image, and that treatment effects extend beyond this region which may be better discriminated with 3D dosimetry.*

Discussion

1. A limitations section should be included.

We have included a Limitations section in the manuscript, within the Discussion section.

2. As discussed in comment above, the authors report that the number of catheters significantly correlated with Vd. This should be discussed further, given that undertreating the tumors can be a major limitation in glioma trials, including those using CED with a single catheter. Will the authors next trial aim to use multiple catheters, regardless of tumor size defined by contrast enhancement?

Please see our response to Question 6 (“Results”).

3. The authors correctly discuss the need to target for CE and NE tissue. Please discuss the strategy for catheter placement for future trials.

Please see our response to Question 6 and Question 7 (“Results”).

4. Pseudoprogression is a critical point the authors bring up. Post-treatment biopsies on catheter removal could be critical to understand treatment response in this context, and are a major advantage to using CED given the direct access to tissue.

We appreciate Reviewer 2's suggestion, however, we did not collect the pretreatment or posttreatment biopsies centrally for further analysis but will plan to do so in the future to better understand our patient population. The total information collected in the study on tumor characterization is provided in the manuscript/supplementary information.

5. Was the 100 Gy absorbed efficacy dose threshold all from cohort 6? Please clarify.

We have revised the Results section to state: *Given the findings from our preclinical studies identifying 100Gy as a threshold for survival benefit²⁰, patients were also dichotomized by absorbed dose of 100 Gy for PFS and OS.*

The median PFS (mPFS) was 4.0 months (95% CI 2.0-6.0 months) with one patient progression-free >11 months (Figure S3a). After dichotomizing by absorbed dose, a significant difference was observed with mPFS, with 2.0 months (95% CI 1.0 to 4.0 months) for <100 Gy and 6.0 months for ≥100 Gy (95% CI 3.0-8.0 months) (Figure S3b). After adjustment for age, baseline ECOG status, baseline volume administered, and baseline tumor volume, PFS increased by 15% for each 10% increase in the percentage of tumor covered (fold change=1.49, 95% CI 1.063, 1.242) and by 19% for each 100 Gy increased in the absorbed dose (fold change=1.187, 95% CI 1.04, 1.356) (Table S4).

The mOS across all cohorts 1 to 6 (n=21) was 11.0 months (95% CI 5.0-17.0 months) with one patient remaining alive 29 months after treatment (Figure 2a). A total of 12/21 patients received ≥100 Gy radiation absorbed dose (Table S8). A significantly longer mOS was observed in those whose tumors received >100 Gy (p<0.001). The mOS was 17.0 months (95% CI 8.0-35 months) for >100 Gy (n=12) compared to 6.0 months (95% CI 1.0-11.0 months) for <100 Gy (n=9) (Figure 2b).

The number of catheters significantly correlated with Vd (r=0.67, p<0.001) and ¹⁸⁶RNL volume infused (r=0.69, p<0.001), and not with tumor coverage (%) (r=0.39, p=0.08), age (r=-0.04, p=0.88), baseline ECOG status (r=-0.41, p=0.07), or absorbed dose (r=0.16, p=0.48).

After adjustment for age, baseline ECOG status, baseline volume administered, and baseline tumor volume, OS for all cohorts increased by 27% for each 10% increase in the percentage of tumor covered (fold change 1.274 95% CI 1.209 to 1.343, p<0.001) and by 31% for each 100 Gy increase in the absorbed dose (fold change 1.312, 95% CI 1.124 to 1.532, p<0.001) (Table S5).

The following reference was used as support:

- Phillips, W. T. et al. Rhenium-186 liposomes as convection-enhanced nanoparticle brachytherapy for treatment of glioblastoma. *Neuro-Oncology* 14, 416–425 (2012).

Methods

1. There needs to be more information about the CED treatment.

a. Clearly state treatment duration as it is unclear in the current form.

We revised the text in the Treatment section (Methods) to state: *The total time of infusion was a function of flow rate, number of catheters used, and total volume infused (Table S7).*

b. Describe the surgical method of catheter placement.

We have revised the text in the Treatment section (Methods) to state: *Once patients were consented, they underwent treatment planning MRI within seven days before drug infusion to evaluate tumor characteristics, including location, structure, shape, and dimension. Based on the treatment planning MRI, patients were further selected for study status. If the tumor had progressed beyond the screening criteria, they were considered a screen failure. Once confirmed to be appropriate for CED, stereotactic treatment planning was centralized and performed by the Principal Investigator (PI) together with the local neurosurgeon. The tumor volume was calculated by the use of iPlan Flow, as was all in silico treatment planning and convection simulation, as has been previously described¹⁸. Briefly, using the iPlan software, the PI would determine the number of catheters and their trajectories, based on the size and location of the tumor, in accordance with pre-set parameters noted below. Early cohorts minimized the number of catheters for safety; once this was determined, the number of catheters used (1-4) was at the treating investigator's discretion based on planning to*

allow for suitable coverage of the enhancing tumor with surrounding T2 FLAIR abnormality. Immediately following a SoC stereotactic biopsy procedure for confirmation of disease progression, BrainLab Flexible CED catheter(s) (7.5 mm tip) were placed, with at least one catheter proceeding along the same needle track as the SoC biopsy. Catheter placement was optimized to avoid ependymal surfaces by >0.5 cm as they offer no resistance to fluid flow and 2.0 cm from the closest sulcus, fissure, resection cavity, or cortical surface. Distal placement of the infusing tip within the tumor was also critical to avoid the pressure gradient from the tumor core to periphery to adjacent brain. Postoperative head CT was performed to evaluate for hematoma or pneumocephalus.

The following reference was used as support:

- Wembacher-Schroeder, E. *et al.* Evaluation of a patient-specific algorithm for predicting distribution for convection-enhanced drug delivery into the brainstem of patients with diffuse intrinsic pontine glioma. *J Neurosurg Pediatr* 28, 34–42 (2021).

c. How was backflow monitored?

Please see our response to Question 8 (“Results”).

2. Why were catheters placed and removed on average 1-2 days before treatment started or ended?

We have revised the text in the Treatment section (Methods) to reflect the placement of catheters, stating: *To allow for fibrin and clot deposition to occur, ¹⁸⁶RNL infusion was performed approximately 24 hours following catheter placement.*

Likewise, we revised the text in the Treatment section (Methods) to reflect the removal of catheters, stating: *CED catheters were removed ~24-48 hours at the direction, timing, and discretion of the study team. This was a function of a number of factors including recommended timing of catheter use^{15,35,36} and optimization of overall procedure time for patients cross with the OR and nuclear medicine.*

The following references were used as support:

- Sampson, J. H. *et al.* Poor drug distribution as a possible explanation for the results of the PRECISE trial. *J Neurosurg* 113, 301–309 (2010).
- Thompson, E. M. *et al.* Recombinant polio–rhinovirus immunotherapy for recurrent paediatric high-grade glioma: a phase 1b trial. *The Lancet Child & Adolescent Health* 7, 471–478 (2023).
- Mueller, S. *et al.* PNOC015: Repeated convection-enhanced delivery of MTX110 (aqueous panobinostat) in children with newly diagnosed diffuse intrinsic pontine glioma. *Neuro-Oncology* 25, 2074–2086 (2023).

3. Please briefly expand upon the in silico treatment planning and convection simulation.

We have revised the text in the Treatment section (Methods) to state: *The tumor volume was calculated by the use of iPlan Flow, as was all in silico treatment planning and convection simulation, as has been previously described¹⁸. Briefly, using the iPlan software, the PI would determine the number of catheters and their trajectories, based on the size and location of the tumor, in accordance with pre-set parameters noted below. Early cohorts minimized the number of catheters for safety; once this was determined, the number of catheters used (1-4) was at the treating investigator’s discretion based on planning to allow for suitable coverage of the enhancing tumor with surrounding T2 FLAIR abnormality. Immediately following a SoC stereotactic biopsy procedure for confirmation of disease progression, BrainLab Flexible CED catheter(s) (7.5 mm tip) were placed, with at least one catheter proceeding along the same needle track as the SoC biopsy. Catheter placement was optimized to avoid ependymal surfaces by >0.5 cm as they offer no resistance to fluid flow and 2.0 cm from the closest sulcus, fissure, resection cavity, or cortical surface. Distal placement of the infusing tip within the tumor was also critical to avoid the pressure gradient from the tumor core to periphery to adjacent brain. Postoperative head CT was performed to evaluate for hematoma or pneumocephalus.*

The following reference was used as support:

- Wembacher-Schroeder, E. *et al.* Evaluation of a patient-specific algorithm for predicting distribution for convection-enhanced drug delivery into the brainstem of patients with diffuse intrinsic pontine glioma. *J Neurosurg Pediatr* 28, 34–42 (2021).

Reviewer #3

We thank Reviewer #3 for their thoughtful consideration of our manuscript and provide the following supporting information to address their open questions.

The authors evaluated the safety, pharmacokinetics, and preliminary efficacy of a new radiotherapeutic product, Rhenium (^{186}Re) Obisbameda (^{186}RNL), in a multicenter, single arm, phase 1 study for the treatment of recurrent glioma. The authors have developed their strategy for re-irradiation in recurrent WHO Grade 3 or 4 glioma using ^{186}Re -nanoliposomes based on pre-clinical data. By directly targeting tumors using convection enhanced delivery (CED), they were able to safely deliver 22.3 mCi in 8.80 ml using up to 4 catheters. The doses of radiation delivered are much higher than those obtainable with conventional external radiotherapy. In this initial analysis, the maximum tolerated dose (MTD) was not reached nor was the dose-limiting toxicity (DLT) and overall survival (OS) was shown to be closely correlated with tumor percent volume coverage by the drug and the dose of radiation absorbed by the tumor. This is a novel approach in an unmet need cancer. It is very appealing to use a tracer that can also be used to image agent distribution in brain. I have several comments and suggestions for revision:

1. Although the method is entirely sound from a neurosurgical point of view and the link with current standards of care is relevant, it would be useful to clarify where this type of treatment can be developed and how it can or cannot be extended everywhere. Also, the cost of such support must be raised through a study of the corresponding economic mode; at least be mentioned.

We have revised the text in the Discussion section to state: *CED also is technically challenging and can be a source of treatment failure even with an efficacious therapeutic. We recognize the infrastructure required, including sophisticated, cost-intensive neuronavigation software and hardware. At present, this technology from companies such as Brainlab or Stryker are currently available in most hospitals with neuro-oncology expertise, and routinely used. For this study and our ongoing Phase 2 trial, treatment planning is performed centrally with participation of the local neurosurgeon.*

2. 21 patients are included in the study, divided into two groups according to the dose deposited ($>100\text{ Gy}$ $n=12$; $<100\text{ Gy}$ $n=9$), demonstrating a survival advantage in patients who received the highest dose: 17 months versus 6 months. These figures are compared to the 8 months obtained for the standard of care. In general, on this subject, it is 8 months but without progression, which is not the case here since it involves recurrences. Can the authors clarify this?

We have tried to clarify the difference between the administered dose and the radiation absorbed dose throughout the manuscript. The study did not prospectively divide patients into radiation absorbed dose categories of $>$ or $<100\text{Gy}$; this was a determined based on post-infusion dosimetry measurements, as described in the Treatment section (Method) and then correlated with overall survival, among others. Regarding the 8 months standard of care following recurrence, the Introduction currently states: *Median overall survival (mOS) with first progression is typically $\sim 7.4\text{-}9.2$ months^{7,8} and with second progression, approximately four months⁹.*

The following references are used as support:

- Friedman, H. S. *et al.* Bevacizumab Alone and in Combination With Irinotecan in Recurrent Glioblastoma. *JCO* 27, 4733–4740 (2009).
- Cloughesy, T. F. *et al.* A randomized controlled phase III study of VB-111 combined with bevacizumab vs bevacizumab monotherapy in patients with recurrent glioblastoma (GLOBE). *Neuro-Oncology* 22, 705–717 (2020).
- Quant, E. C. *et al.* Role of a second chemotherapy in recurrent malignant glioma patients who progress on bevacizumab. *Neuro-Oncology* 11, 550–555 (2009).

3. The choice of ^{186}Re (with a half-life of ~ 89.2 hours) is made here. It is a reactor-produced isotope that can also be generated by cyclotron. Can the authors comment on the interest, advantages and disadvantages of this product. On this subject, the constraint for clinicians and patients (37 days for zero radioactivity) is not mentioned? Could the same type of work have been done with ^{188}Re which is a generator product and has a half-life of ~ 16.9 hours (total decay in 7 days)?

We have revised the text in the introduction to state: *^{186}Re was preferentially chosen over ^{188}Re , which is generator produced, due to its favorable mean path length of 1.8 mm (compared to ^{188}Re 's 3.5 mm) to better avoid healthy tissue penetration for lower normal tissue toxicity²². Additionally, the authors previously reported on ^{186}RNL preclinically²⁰.*

The following reference was used as support:

- Uccelli, L. *et al.* Rhenium Radioisotopes for Medicine, a Focus on Production and Applications. *Molecules* 27, 5283 (2022).
- Phillips, W. T. *et al.* Rhenium-186 liposomes as convection-enhanced nanoparticle brachytherapy for treatment of glioblastoma. *Neuro-Oncology* 14, 416–425 (2012).

4. In link with the above, it is essential to discuss the choice of the radiopharmaceutical (¹⁸⁶Re-nanoliposome). In this regard, it is important to highlight recent preclinical developments more broadly on the keywords: radiotherapy, rhenium, nanoparticle, glioma (eg. DOI: 10.7150/thno.19403; DOI: 10.1016/j.biomaterials.2011.05.067).

We have revised the introduction to incorporate the key publications that informed the authors prior to study start, now stating: *Prior to study start we searched PubMed from any time up to December 1, 2013 for clinical therapeutic studies using ¹⁸⁶Re to treat glioma using the search terms “(¹⁸⁶Re[ALL FIELDS] AND glioma[ALL FIELDS]) and (rhenium-186[ALL FIELDS] AND glioma[ALL FIELDS]). The search gave four publications^{23–26}. Intramedullary cystic spinal cord pilocytic astrocytoma was managed with minor side effects using ¹⁸⁶Re intracavitary irradiation, with stabilization of the cyst and neurological deficit improvement²³. Combination treatment for pilocytic astrocytoma using ¹⁸⁶Re delivered 400 Gy to the cyst wall and resulted in progressive cyst disappearance and mural nodule retraction²⁶. A fibrin glue of ¹⁸⁸Re and ¹⁸⁶Re bound in microspheres was used post-tumor resection in a 9L-glioblastoma rat model. 60% of treated animals survived 36 days, compared to control animals (17 ± 3 days)²⁴. Lastly, intracavitary ¹⁸⁶Re application was used in six cases of cystic craniopharyngiomas but abandoned because of cyst recurrence and leakage²⁵.*

The following references were used as support:

- Colnat-Coulbois, S., Klein, O., Braun, M., Thouvenot, P. & Marchal, J.-C. Management of Intramedullary Cystic Pilocytic Astrocytoma With Rhenium-186 Intracavitary Irradiation: Case Report. *Neurosurgery* 66, E1023–E1024 (2010).
- Häfeli, U. O., Pauer, G. J., Unnithan, J. & Prayson, R. A. Fibrin glue system for adjuvant brachytherapy of brain tumors with ¹⁸⁸Re and ¹⁸⁶Re-labeled microspheres. *Eur J Pharm Biopharm* 65, 282–288 (2007).
- Gahbauer, H. *et al.* Combined use of stereotaxic CT and angiography for brain biopsies and stereotaxic irradiation. *AJNR Am J Neuroradiol* 4, 715–718 (1983).
- Proust, F. *et al.* [Combination treatment for pilocytic astrocytoma: stereotaxic radiosurgery and endocavitary radiotherapy]. *Neurochirurgie* 44, 50–54 (1998).

5. The radiation absorbed dose calculation uses the MIRDose algorithm which assumes 100% of the beta-radiation dose has deposited locoregionally. It is not perfectly clear how the dose distribution study attest to this point (volume of ¹⁸⁶Re distributed in the brain and in the tumor, percentage of tumor volume treated, radiation absorbed dose to the tumor, and whole-body normal organ doses)?

The MIRDose algorithm has been used as a standard for clinically calculating radiation absorbed doses in normal organs. The corresponding OLINDA software used is an FDA-approved software for normal organ radiation absorbed dose calculation. The assumption is an embedded component of the MIRDose algorithm. We therefore consider that the assumption used by MIRDose is based on the following: 1) there is only < 1mm-5mm range in the tissue with beta-particle therapeutic radiation depending on the energies of the beta-particle (there is only 1.8 mm average range for the beta-radiation from ¹⁸⁶Re); 2) the nuclear imagers for clinical patients has poorer image resolution (~ 8 mm to 1 cm) than the radiation range; and 3) for the evaluation of normal organ effect, more micro-scale radiation dose calculation may not be necessary, if the normal organ doses are not high. Meanwhile, we understood the evaluation of radiation absorbed dose distribution, especially the coverage of radiation therapy, can be very important. Accordingly, the specific methodology for the evaluation of locoregional distribution of therapeutic agent at the tumor region has been developed. The data have been evaluated to investigate its relationship with therapy effect, like patient survival, which has also been described in the manuscript.

6. Regarding, MGMT status, it has indeed been demonstrated that it impacts the effectiveness of temozolomide (TMZ), which however was not used in this study. Does the author have any evidence that it can define ¹⁸⁶Re-radioresponse which will give echoes of how the data produced here on this subject could be further used (larger cohorts) in terms of prognosis feature.

We thank the reviewer for the insightful question. We have requested archival specimens which can be analyzed retrospectively to identify prognostic factors. DNA damage response pathways are of keen interest, especially the ATR-CHK1 pathway, due to stabilizing replication forks encountering oxidatively-damaged DNA or DSBs. Inhibiting replication fork arrest and then restart after repair would lead to synergy with the Rhenium Obisbameda. At present, the number of data points is low which precludes meaningful discovery. As we acquire additional survival data and archival specimens in the phase 2 trial we plan to analyze this further. Additionally, in an ongoing study of rhenium obisbameda for leptomeningeal metastases we are able to collect tumor cells at various time points after treatment. This is allowing us to profile response and identify DDR alterations that may correlate with response. We look forward to presenting those findings at a later date.

7. Although this reviewer is not a statistician, it seems important to include the contribution of an expert in the field in this study. For example, there is no mention of calculated sample size, hazard ratio (HR) or randomization. Can you clarify this point?

A statistician contributed to the design of the study and its analysis. We have revised the text in the Statistics section (Methods) to state: *Discrete outcomes were summarized with frequencies and percentages, and continuously distributed outcomes with the mean and standard deviation (SD). Absorbed dose was dichotomized to “ ≥ 100 Gy and < 100 Gy” as defined by absorbed dose ≥ 100 and absorbed dose < 100 , respectively. The significance of variation in the mean and cohort was assessed with analysis of variance. The significance of associations between categorical outcome and cohort was assessed with chi-square and Fisher’s exact tests. Variation in survival with category of absorbed dose was assessed, without covariate adjustment, with log rank tests and with accelerated failure time (AFT) models with adjustment for covariates; a lognormal error term was assumed. All measurements were made on individual patients. No repeated measures within patient are shown. Covariates were used in the AFT models and are specified in the summary tables. Except for the AFT models, where the logarithm of survival (PFS and OS) is modelled in terms of covariates, all statistical analyses were done in original units. Measures of central tendency such as the mean and median are specified in the tables. Fold change was defined as the antilog of the beta coefficient in the AFT model. No Bayesian methods and no hierarchical models were used. The sample sizes were specified in accordance with 3+3 methodology; power calculations were not made and were not used. Corrections for multiple comparisons were not applied. All significance testing was 2-sided with a significance level of 5%. SAS Version 9.4 (SAS Institute, Cary NC) was used.*

Reviewer #4

We thank Reviewer #4 for their thoughtful consideration of our manuscript and provide the following supporting information to address their open questions.

The authors reported the results of the Phase I dose escalation trial NCT01906385 that evaluated a single dose of 186RNL administered through a convection enhanced delivery catheter in participants with recurrent Glioma (GBM). The study employed a modified 3+3 Fibonacci dose escalation method. The maximum tolerated dose (MTD) was not reached, and overall survival was found to correlate with both the percentage of tumor volume coverage by the drug and the radiation absorbed dose to the tumor. The manuscript is well-written. We suggest adding a few more details on the study design and results for rigor and reproducibility, and for interpretation of the results, as recommended by the Dose Finding CONSORT extension (DEFINE-CONSORT; BMJ 2023 Oct 20;383:e076387. doi: 10.1136/bmj-2023-076387.)

1. The definitions of the endpoints should be included and/or clarified:
- The DLT definition, even if it follows the standard definition based on grade 3 of the CTCAE scale, should be clearly stated, including the DLT evaluation window.

We have revised the text in the new Safety section to state: *For this study, a dose limiting toxicity (DLT) was defined as grade 3 or greater acute CNS toxicity attributable to the study intervention which persists for 96 hours or more (see below discussion of delayed events) OR grade 3 or greater non-CNS toxicity which is attributable to the study intervention, as per the grading scale of the CTCAE v4.0²⁹. The standard DLT window was 28 days following treatment. Additionally, given the possibility for radiation effects outside of the standard 28-day DLT window, additional consideration was given that extended the DLT evaluation period for CNS toxicity to 90 days between successive cohorts. Lastly, if a patient within a cohort experienced a CNS toxicity that would be defined as dose limiting, the entire cohort would complete 90 days evaluation before the successive cohort would commence.*

The following reference was used in support:

- National Institute of Cancer (NCI). Common Terminology Criteria for Adverse Events (CTCAE), v4.0. <http://evs.nci.nih.gov/ftp1/CTCAE/About.html>.

b. The time point for the absorbed dose could be stated more clearly.

We currently note in the Treatment section the following: *Finally, cumulated radioactivity (\bar{A}) in the tumor within 192 hours of infusion was calculated and used to determine the mean radiation absorbed dose in the tumor.* The radiation absorbed dose is a cumulation of data over a time period (start time through 192 hours), and not taken from a specific time point.

2. The timepoint at which the volume of distribution was obtained can also be clarified.

As noted above, the volume of distribution is determined by using a cumulation of data over 192 hours post drug infusion.

3. While the paper mentions that the 3+3 design is used, it does not clearly state the dose-escalation decisions and the definition of MTD. For example,

a. Whether other endpoints were considered for the dose escalation (e.g., pharmacokinetic measures).

The primary objective of the study was to determine the maximum tolerated dose of ^{186}RnL by convection enhanced delivery (CED) at the time of planned stereotactic biopsy, when necessary, as standard of care. The secondary objectives were to assess the safety of single dose ^{186}RnL by CED; assess the dose distribution of ^{186}RnL by CED; determine the overall response rate by Radiographic Assessment in Neuro-Oncology (RANO) criteria following ^{186}RnL treatment; determine disease specific progression-free survival after ^{186}RnL treatment; and determine overall survival (OS) after ^{186}RnL treatment. This is also noted in the Introduction section, where it states: *Following these studies, we initiated ReSPECT-GBM, a multicenter, sequential cohort, open-label, volume and dose escalation study of the safety, tolerability, and distribution of a single dose of ^{186}RnL given by CED for recurrent glioma. The primary objective was to determine the maximum tolerated dose (MTD) of ^{186}RnL by CED in patients with recurrent glioma. Secondary objectives included the assessment of the safety and tolerability of a single dose ^{186}RnL , the dose distribution of ^{186}RnL by imaging, the overall response rate (ORR), disease-specific progression-free survival (PFS), and OS. A graphical summary of the trial design and its outcomes is included in Figure S4.*

b. The 3+3 would not have 6 patients in cohort 6 in the absence of DLT. What was the rationale for adding three more patients in cohort 6 (with increasing maximum flow rate)? Was this pre-specified in the protocol?

We have revised the text in the Discussion section to state: *Following interim analysis of cohort 6, we observed a plateau from cohort 5 to cohort 6 in both dose distribution and absorbed dose. As a result of this finding, the recommendation of the Data Safety Monitoring Board (DSMB) was to expand cohort 6 to confirm tolerability of 22.3mCi administered dose in 8.8mL infusate volume as the Phase 2 recommended dose (RP2D) to target both bulk disease and adjacent microscopic disease in patients with one recurrence and tumor sizes of 20 cm³ or less. Once completed, and without the MTD being reached, all cohort data was analyzed with recommendation from the DSMB to proceed with Phase 2 utilizing this dose.* This was done under a protocol amendment and not pre-specified.

4. The dose levels to be evaluated should be specified in the methods along with the rationale for them and the starting dose. It is not clear if:

a. The Fibonacci was used on RnL activity and the infused volume was calculated from that.

The modified Fibonacci was based on the administered dose. We have revised the text in the Demographics section (Results) to state: *The study used a modified 3+3 dose escalation²⁷, with increased in total radioactivity by doubling from 1 mCi in Cohort 1 to 8 mCi in Cohort 4, followed by a 66% increase in cohorts 5 and 6. The administered dose range was 1 mCi in a volume of 0.66 mL through 22.3 mCi in 8.80 mL (Table 1).* It should be emphasized that the safety of each cohort was review by an independent Data Safet Monitoring Board who made recommendations on the dose escalation. Given the lack of significant AEs in the first 4 cohorts, the DSMB, dose doubling was recommended.

The following reference was used in support:

- Le Tourneau, C., Lee, J. J. & Siu, L. L. Dose Escalation Methods in Phase I Cancer Clinical Trials. *JNCI: Journal of the National Cancer Institute* 101, 708–720 (2009).

b. The methods should also clarify the relationship between the assigned and absorbed dose.

We have tried to clarify the difference between the administered dose and the radiation absorbed dose throughout the manuscript.

5. Whether a sentinel patient was included or not.

In the context of a sentinel patient as a marker of toxicity, we did not. As mentioned, the toxicity was modest and as such there was not specific toxicity to follow in other patients.

6. We appreciated the results presented both by cohort and by the total number of trial patients. In general, the results are well-reported and discussed. Further clarifications on:

- a. The number of catheters used by cohort and in total should be reported as this was left to the discretion of the treating physician and it is not clear if it differed by cohort.

We have revised the text in the Treatment section (Methods) to state: *Once patients were consented, they underwent treatment planning MRI within seven days before drug infusion to evaluate tumor characteristics, including location, structure, shape, and dimension. Based on the treatment planning MRI, patients were further selected for study status. If the tumor had progressed beyond the screening criteria, they were considered a screen failure. Once confirmed to be appropriate for CED, stereotactic treatment planning was centralized and performed by the Principal Investigator (PI) together with the local neurosurgeon. The tumor volume was calculated by the use of iPlan Flow, as was all in silico treatment planning and convection simulation, as has been previously described¹⁸. Briefly, using the iPlan software, the PI would determine the number of catheters and their trajectories, based on the size and location of the tumor, in accordance with pre-set parameters noted below. Early cohorts minimized the number of catheters for safety; once this was determined, the number of catheters used (1-4) was at the treating investigator's discretion based on planning to allow for suitable coverage of the enhancing tumor with surrounding T2 FLAIR abnormality. Immediately following a SoC stereotactic biopsy procedure for confirmation of disease progression, BrainLab Flexible CED catheter(s) (7.5 mm tip) were placed, with at least one catheter proceeding along the same needle track as the SoC biopsy. Catheter placement was optimized to avoid ependymal surfaces by >0.5 cm as they offer no resistance to fluid flow and 2.0 cm from the closest sulcus, fissure, resection cavity, or cortical surface. Distal placement of the infusing tip within the tumor was also critical to avoid the pressure gradient from the tumor core to periphery to adjacent brain. Postoperative head CT was performed to evaluate for hematoma or pneumocephalus.*

To allow for fibrin and clot deposition to occur, ¹⁸⁶RNL infusion was performed approximately 24 hours following catheter placement. ¹⁸⁶RNL was manufactured as previously described²⁰. Super-saturated potassium iodide (SSKI, 600 mg) was administered by mouth with water or juice prior to infusion. The infusion rate started at 1 μ l/min and stepped to 20 μ l/min (Table S6) using a syringe pump (Medfusion 3500 and 4000, Adepto Medical, Kansas City, MO). The total time of infusion was a function of flow rate, number of catheters used, and total volume infused (Table S6, Table S7, Table 1). During infusion, planar and tomographic images were collected using a dual-detector SPECT/CT camera. A sealed vial with known ¹⁸⁶RNL was positioned next to the patient at each time of image acquisition for radioactivity quantification. Dynamic images were acquired via real-time persistence scope for evaluation of focal accumulation of activity at the assumed tip of the catheter(s). When activity was observed to accumulate focally, that time was designated as the beginning of the planned therapeutic volume infusion to correct for dead space in the catheter line. Catheters deemed unsatisfactory (backflow along the catheter or spillage into adjacent CSF space) were stopped and the remaining volume switched to the remaining catheters.

To minimize backflow concerns³¹, we utilized the Brainlab catheter, which is both flexible and has a “step design” at the tip, avoided resection cavities and pial surfaces, and employed a slow ramping of infusion rates (Table S6). Furthermore, backflow was monitored during infusion using planar imaging as mentioned above, allowing changes to be made in real-time to ensure the planned administered dose was delivered in full.

The following reference was used as support:

- Wembacher-Schroeder, E. *et al.* Evaluation of a patient-specific algorithm for predicting distribution for convection-enhanced drug delivery into the brainstem of patients with diffuse intrinsic pontine glioma. *J Neurosurg Pediatr* **28**, 34–42 (2021).

b. Report the AE by cohort using patient as unit of analysis to know what proportion of patients experienced those grades of AE in each cohort. The frequency of events using events are unit (Tables 4 and 5) can be supplemental.

We have revised the AE tables accordingly.

c. Summary statistics on the absorbed doses per cohort.

We have provided a table for absorbed doses per patient/cohort (Table S8) and a figure of the summary per cohort (Figure 1).

d. In Figure 1 (page 6), while the median Vd in cohort 6 is higher compared to all other cohorts, the maximum Vd is observed in a patient in cohort 5. Do the authors have an explanation for this?

We have revised the text in the Discussion section to state: *Following interim analysis of cohort 6, we observed a plateau from cohort 5 to cohort 6 in both dose distribution and absorbed dose. As a result of this finding, the recommendation of the Data Safety Monitoring Board (DSMB) was to expand cohort 6 to confirm tolerability of 22.3mCi administered dose in 8.8mL infusate volume as the Phase 2 recommended dose (RP2D) to target both bulk disease and adjacent microscopic disease in patients with one recurrence and tumor sizes of 20 cm³ or less. Once completed, and without the MTD being reached, all cohort data was analyzed with recommendation from the DSMB to proceed with Phase 2 utilizing this dose.*

e. For survival analysis, was the dichotomization (<100 Gy or not) decided before the study onset, or is it a post-hoc analysis? Why was the survival analysis not done by assigned dose/cohort like ORR, but by absorbed dose?

We have tried to clarify the difference between the dose administered and radiation absorbed dose throughout the manuscript. While the study prospectively assigned patients to cohorts differing in administered dose, the study did not prospectively divide patients into radiation absorbed dose categories of > or <100Gy. It was not possible to know *a priori* what absorbed dose would be for a given administered dose as this is highly dependent on both distribution and retention. Indeed, this was a determined on a case by case basis using post-infusion dosimetry measurements, as described in the Treatment section and then correlated with overall survival, among others. Survival analysis by Cohort – or administered dose – did not show any correlation with overall survival, as the enhancing tumor volume was required to be within the treatment field volume for a respective cohort, per the inclusion criteria. We therefore presented the survival data as we did, for all patients and the dichotomization, which we saw in our preclinical studies. We have revised the Results section to state: *Given finding from our preclinical studies identifying 100Gy as a threshold for survival benefit²⁰, patients were dichotomized by absorbed dose of 100 Gy.* Likewise, in the Introduction section, we state: *Furthermore, >100 Gy of ¹⁸⁶RNL was shown to eradicate grafted tumors and prolong overall survival, without clinical or microscopic evidence of toxicity²⁰.*

The following reference was used as support:

- Phillips, W. T. *et al.* Rhenium-186 liposomes as convection-enhanced nanoparticle brachytherapy for treatment of glioblastoma. *Neuro-Oncology* **14**, 416–425 (2012).

f. For Table S4, did the authors examine the correlation between “Ratio of Treated to Total Tumor Volume”, “Total Dose in Distribution Volume”, and “Volume Administered” for the “Progression-Free Survival Accelerated Failure Time Model”? It is not clear why both of this need to be in the model.

We thank the reviewer for the question, but we are not completely sure of the intended meaning of the question. We believe this question to mean that some of the variables assessed are redundant. The first variable was to assess coverage independent of the dose administered, the second to assess the absorbed dose, and the third an ITT based on what was given rather than achieved.

- g. The trial accrued more males than females and mostly white patients; this could be discussed in terms of generalizability.

We thank the reviewer and note that the study population was reflective of the patient population in the centers participating. We have revised the text in a new Sex and Gender section to address the information regarding male/female accrual to state: *Sex and gender, as determined based on self-reporting, was not a factor in patient inclusion criteria, nor were there sufficient numbers of patients of either gender to make a meaningful sex- and gender-based conclusions in the study; as such, sex- and gender-based analyses have not been reported here.* Based on the small number of patients, we also extend this same rationale to race/ethnicity.

Reviewer #5

We thank the reviewer for their co-review and have provided responses to the comments of Reviewers 1-4.

Nature Communication Manuscript Response

Title: Convection Enhanced Delivery of Rhenium (^{186}Re) Obisbameda in Recurrent Glioma: a multicenter, single arm, phase 1 clinical trial

Corresponding Author: Andrew Brenner, MD, PhD

Manuscript Revision: NCOMMS-24-25017-B

Reviewer #1

We thank Reviewer #1 for their thoughtful consideration of our revised manuscript and our response.

Reviewer #2

We thank Reviewer #2 for their thoughtful consideration of our revised manuscript and our response.

Reviewer #3

We thank Reviewer #3 for their thoughtful consideration of our revised manuscript and our response and provide the following supporting information to address their open questions.

1. Although the authors have generally done their best to improve the manuscript, question number 4 of this reviewer was not addressed from the angle necessary to understand the interest of the radiopharmaceutical (radiolabeled nanoliposome or rhenium-loaded nanoliposome). Major: The advantages of the vector were not presented as a possible alternative considering other colloids in the literature (biodistribution profiles, stability, safety). The four reviews cited are not directly related to the choice of the nanoliposome and its performances. They mainly consider static implants tested in the clinic but not dynamic nanovectors also in development. Why this one and not another? Would another colloid have achieved the same result? What about what is available from preclinical to clinical?

We appreciate that our answer, and the concomitant updates to the manuscript, for Question 4, “*In link with the above, it is essential to discuss the choice of the radiopharmaceutical (^{186}Re -nanoliposome). In this regard, it is important to highlight recent preclinical developments more broadly on the keywords: radiotherapy, rhenium, nanoparticle, glioma (eg. DOI: 10.7150/thno.19403; DOI: 10.1016/j.biomaterials.2011.05.067).*” sufficiently addressed the published state of the art prior to beginning the trial for rhenium-based compounds used for glioma. Regarding “Why nanoliposomes?” or “Why this kind of nanoliposome?” we have updated the manuscript to provide some additional context.

We have revised the references and text in the Introduction to state: *Liposomes are spherical, self-assembling vesicles made up of one or more naturally occurring lipid bilayers in a central compartment, which makes them ideal candidates as delivery vehicles for small molecules, proteins, nucleic acids, and imaging agents for therapeutic and diagnostic use¹⁹⁻²¹. Although nanoliposomes were previously shown to be amenable to loading of diagnostic imaging agents, including $^{99\text{m}}\text{Tc}$, ^{67}Ga , and ^{111}In ²⁰, the mechanisms of labeling were not transferrable to all isotopes. We previously demonstrated the ability to load nanoliposomes with rhenium-186 (^{186}Re , half-life ~89.2 hours) using a custom lipophilic molecule, N,N-bis(2-mercaptoethyl)-N',N'-diethylethylene diamine (BMEDA)²²⁻²⁵ and encapsulation results in improved bioavailability and distribution of chelated- ^{186}Re in tumor tissue²⁶. The final product – Rhenium (^{186}Re) Obisbameda (^{186}RNL) – can be applied to both nuclear imaging and targeted radionuclide therapy.*

The following references were used as support:

- Liu, C.-M. *et al.* Preliminary evaluation of acute toxicity of ^{188}Re -BMEDA-liposome in rats. *J. Appl. Toxicol.* 30, 680–687 (2010).
- Low, H. Y. *et al.* Radiolabeled Liposomes for Nuclear Imaging Probes. *Molecules* 28, 3798 (2023).
- Dymek, M. & Sikora, E. Liposomes as biocompatible and smart delivery systems - the current state. *Adv Colloid Interface Sci* 309, 102757 (2022).
- Bao, A. *et al.* A Novel Liposome Radiolabeling Method Using $^{99\text{m}}\text{Tc}$ -“SNS/S” Complexes: In Vitro and In Vivo Evaluation. *Journal of Pharmaceutical Sciences* 92, 1893–1904 (2003).
- Phillips, W. T. *et al.* Rhenium-186 liposomes as convection-enhanced nanoparticle brachytherapy for treatment of glioblastoma. *Neuro-Oncology* 14, 416–425 (2012).

- Bao, A., Goins, B., Klipper, R., Negrete, G. & Phillips, W. T. 186Re-liposome labeling using 186Re-SNS/S complexes: in vitro stability, imaging, and biodistribution in rats. *J Nucl Med* 44, 1992–1999 (2003).
- Goins, B., Bao, A. & Phillips, W. T. Techniques for Loading Technetium-99m and Rhenium-186/188 Radionuclides into Pre-formed Liposomes for Diagnostic Imaging and Radionuclide Therapy. in *Liposomes* (ed. Weissig, V.) vol. 606 469–491 (Humana Press, Totowa, NJ, 2010).
- French, J. T. et al. Interventional Therapy of Head and Neck Cancer with Lipid Nanoparticle-carried Rhenium 186 Radionuclide. *Journal of Vascular and Interventional Radiology* 21, 1271–1279 (2010).

Given the constraints of the manuscript, the previously published papers on the labeling methods (listed above), and the focus of the manuscript on the Phase 1 trial, we could not include a comprehensive literature review of carrier technology at large in this article but have added a more appropriate review article and additional clarification to meet the reviewers concern. We hope that the additional supporting information, in concert with the previous changes to the manuscript to better explain the choice of ¹⁸⁶Re and its use in glioma, provides appropriate context of drug design at the time of study initiation.

Reviewer #4

We thank Reviewer #4 for their thoughtful consideration of our revised manuscript and our response and provide the following supporting information to address their open questions.

1. Based on the revision, “Given the lack of significant AEs in the first 4 cohorts, the DSMB, dose doubling was recommended.” The manuscript should state that cohorts 5 and 6 were added later after review of data from first 4 cohorts as doses in phase 1 trials are generally pre-specified for rigor and reproducibility. Please specify the set of doses that were specified at the start of the trial in the treatment section and those that were added after review and the rationale for those dose selection.

We apologize for the confusion caused by our prior response. The dose escalation scheme was determined prior to study start; however, the decision to escalate to the next Cohort was dependent on the determination of the DSMB’s review of any given Cohort. We did not deviate from the initial plan.

Cohort	Administered Dose (mCi)	Percent Change
1	1.0	NA
2	2.0	100%
3	4.0	100%
4	8.0	100%
5	13.4	67%
6	22.3	67%

We have revised the text in the Results, Demographics section to state: *The study used a modified 3+3 dose escalation³¹, with increased in total radioactivity by doubling from 1 mCi in Cohort 1 to 8 mCi in Cohort 4, followed by a 67% increase in Cohorts 5 and 6. All administered doses were determined prior to study start, with dose escalation to a subsequent Cohort following confirmation by the Data Safety Monitoring Board (DSMB), after a review of all safety data. The administered dose range was 1 mCi in a volume of 0.66 mL through 22.3 mCi in 8.80 mL (Table 1).*

The following reference was used as support:

- Le Tourneau, C., Lee, J. J. & Siu, L. L. Dose Escalation Methods in Phase I Cancer Clinical Trials. *JNCI: Journal of the National Cancer Institute* 101, 708–720 (2009).

2. Table 2 for the AE is still by events not patients. The unit of analysis is still events to have 24 mild events in the first cohort with 3 patients. Please specify the number of AE out of the 3 patients taking the maximum grade of AE.

We have updated all AE tables accordingly with number per Cohort specified.

3. The variability in the absorbed dose is of concern (Table S8) and not appropriate to summarize at mean +/- SEM in Figure 1. Given the small sample size and large variability per cohort it is best to graph all points. Please explain the poor correlation between administered dose and absorbed dose. Was this evaluated? How reproducible is the absorbed dose?
4. It is important to note that survival was not associated with administered dose and only with absorbed dose in the paper. This is very important and should be clearly explained and discussed. Absorbed dose is an outcome which could be dependent on many factors.

For Questions 3 and 4: We apologize for any confusion in the manuscript regarding administered dose and absorbed dose. Administered dose (mCi) was reproducibly manufactured and quality controlled as outlined in the Chemistry, Manufacturing, and Controls (CMC) section of the IND (and described in brief in Methods, ¹⁸⁶RNL Manufacturing) to ensure that the radioactivity delivered to the patient was standardized per Cohort, i.e., 6.6 mCi was delivered to each patient in Cohort 1.

There were some clear factors that would result in non-proportional increase in absorbed dose with administered dose. First, due to FDA clinical reviewers concerns about high doses to larger tumors, a proportional increase in volume was included in the dose escalation with activity leading to a stable concentration administered through Cohort 4. This means that as we increase dose, the distribution increased proportionately. The volume was held steady at Cohort 5, which resulted in a proportionate increase in absorbed dose. Further, early Cohorts (particularly Cohorts 2 and 3) included patients with progression on bevacizumab. As discussed in the manuscript, bevacizumab impacted CED and resulted in poor absorbed doses, leading to it being added as an exclusion criterion. Finally, as we gained experience with the safety and distribution, additional catheters were used to aid in distribution that adds additional variability. We captured this information in the Methods, Treatment section:

The tumor volume was calculated by the use of iPlan Flow, as was all in silico treatment planning and convection simulation, as has been previously described¹⁸. Briefly, using the iPlan software, the PI would determine the number of catheters and their trajectories, based on the size and location of the tumor, in accordance with pre-set parameters noted below. Early cohorts minimized the number of catheters for safety and to characterize the distribution; once this was determined, the number of catheters used (1-4) was based on planning to allow for suitable coverage of the enhancing tumor with surrounding T2 FLAIR abnormality. Immediately following a SoC stereotactic biopsy procedure for confirmation of disease progression, BrainLab Flexible CED catheter(s) (7.5 mm tip) were placed, with at least one catheter proceeding along the same needle track as the SoC biopsy. Catheter placement was optimized to avoid ependymal surfaces by >0.5 cm as they offer no resistance to fluid flow and 2.0 cm from the closest sulcus, fissure, resection cavity, or cortical surface. Distal placement of the infusing tip within the tumor was also critical to avoid the pressure gradient from the tumor core to periphery to adjacent brain.

Absorbed dose to the tumor per Cohort, therefore, was a function of a number of factors outside the administered dose, which was constant per Cohort. In general, we noted that where pre-treatment planning could be optimally maximized to deliver the administered dose to the tumor and only the tumor the absorbed dose to the tumor was high. In instances where the absorbed dose to the tumor was lower than another patient in the same Cohort, it could be a function of drug diffusing away to an area of low resistance like a nearby resection cavity, non-optimal catheter placement due to the safety planning rules in the protocol, or a tumor that was simply more maximally covered by the volume of distribution because it was smaller than another tumor. In other words, personalized pre-treatment planning is a critical component of Rhenium (¹⁸⁶Re) Obisbameda for the treatment of recurrent glioblastoma.

Table S8 was provided to provide the intra-cohort variability – based on what is noted above – and Figure 1 describes the relationship between the volume of distribution and the absorbed dose.

Lastly, we attempted to describe these issues in the Discussion section:

In this study the maximum absorbed dose was ~740 Gy. Both preclinical dose finding studies and statistical modeling of the clinical data with long rank testing suggested a dose effect of greater than 100Gy. When this dose threshold was applied, a significant marginal correlation was observed between dose, both in its binary (<100 Gy, >100 Gy) and continuous form, and OS. A similar marginal correlation was also observed between percent tumor coverage and OS. Furthermore, these correlations remained statistically significant after adjustment for baseline factors in multivariate models. While further

assessment of the data using a combination of perfusion and delayed contrast imaging is ongoing, we have selected a primary endpoint of OS for our Phase 2 study, currently underway.

Although we found a correlation between the number of catheters, volume of distribution, and the administered volume, we did not observe a correlation between number of catheters and percent tumor coverage. There are a number of confounders that may impact correlation between number of catheters and percent tumor coverage, particularly that treatment planning for larger tumors likely would drive a decision to place a higher number of catheters, but that the inherent challenges of larger tumors would make coverage less successful. Coverage of greater than 70% appeared to be associated with benefit (Figure S1b), but this is likely a factor of the fixed thresholding method used relative to maximum voxel radioactivity (count) of each SPECT image, and that treatment effects extend beyond this region which may be better discriminated with 3D dosimetry.

Several limitations were present in this study which will require further evaluation. First, since this is a first-in-human dose escalation study, including varying doses, administration rates, number of catheters, and tumor sizes, any results should be taken as preliminary and will require further validation. From a dose distribution perspective, SPECT as an imaging modality to be used for dosimetry and absorbed dose quantification has inherent technological limitations when addressing CNS tumors based on camera resolution, voxel size, and brain anatomy. Further, distribution was estimated based on activity rather than isodose lines. We have since begun performing three-dimensional dosimetry which can define treatment volumes based upon absorbed dose isodose lines and will better define biologically relevant distribution. CED also is technically challenging and can be a source of treatment failure even with an efficacious therapeutic. We recognize the infrastructure required, including sophisticated, cost-intensive neuronavigation software and hardware. At present, this technology from companies such as Brainlab or Stryker are currently available in most hospitals with neuro-oncology expertise, and routinely used. For this study and our ongoing Phase 2 trial, treatment planning is performed centrally with participation of the local neurosurgeon. As reported, following early cohorts, multiple catheters were used whenever possible to maximize distribution, minimize convection time, improve coverage of irregular shaped tumors, and to provide redundancy in case of catheter failure. The ability to visualize the delivery of the therapeutic directly at the time of administration is a unique advantage of CED with a radiopharmaceutical, and further analysis of the entire data set is ongoing and will be reported. Finally, determining response can be challenging in CNS tumors, particularly with radiation which can result in pseudo-progression. To better characterize response, we are performing advanced imaging modalities including Delta T1 subtraction mapping, DSC perfusion, and delayed contrast mapping (TRAMs). In preliminary parametric analyses we observed a statistically significant difference in repeated pair measurements with untreated tumor volume significantly increased relative to treated tumor volume ($p < 0.0001$) suggesting that progression arises outside the treated volume.

Reviewers #5, #6, and #7

We thank Reviewers #5, #6, and #7 for co-reviewing the revised manuscript and our response.

Original Response for Reviewers 3 and 4 for Reference

Reviewer #3

We thank Reviewer #3 for their thoughtful consideration of our manuscript and provide the following supporting information to address their open questions.

The authors evaluated the safety, pharmacokinetics, and preliminary efficacy of a new radiotherapeutic product, Rhenium (186Re) Obisbameda (186RNL), in a multicenter, single arm, phase 1 study for the treatment of recurrent glioma. The authors have developed their strategy for re-irradiation in recurrent WHO Grade 3 or 4 glioma using 186Re-nanoliposomes based on pre-clinical data. By directly targeting tumors using convection enhanced delivery (CED), they were able to safely deliver 22.3 mCi in 8.80 ml using up to 4 catheters. The doses of radiation delivered are much higher than those obtainable with conventional external radiotherapy. In this initial analysis, the maximum tolerated dose (MTD) was not reached nor was the dose-limiting toxicity (DLT) and overall survival (OS) was shown to be closely correlated with tumor percent volume coverage by the drug and the dose of radiation absorbed by the tumor. This is a novel approach in an unmet need cancer. It is very appealing to use a tracer that can also be used to image agent distribution in brain. I have several comments and suggestions for revision:

1. Although the method is entirely sound from a neurosurgical point of view and the link with current standards of care is relevant, it would be useful to clarify where this type of treatment can be developed and how it can or cannot be extended everywhere. Also, the cost of such support must be raised through a study of the corresponding economic mode; at least be mentioned.

We have revised the text in the Discussion section to state: *CED also is technically challenging and can be a source of treatment failure even with an efficacious therapeutic. We recognize the infrastructure required, including sophisticated, cost-intensive neuronavigation software and hardware. At present, this technology from companies such as Brainlab or Stryker are currently available in most hospitals with neuro-oncology expertise, and routinely used. For this study and our ongoing Phase 2 trial, treatment planning is performed centrally with participation of the local neurosurgeon.*

2. 21 patients are included in the study, divided into two groups according to the dose deposited (>100 Gy n=12; <100 Gy n=9), demonstrating a survival advantage in patients who received the highest dose: 17 months versus 6 months. These figures are compared to the 8 months obtained for the standard of care. In general, on this subject, it is 8 months but without progression, which is not the case here since it involves recurrences. Can the authors clarify this?

We have tried to clarify the difference between the administered dose and the radiation absorbed dose throughout the manuscript. The study did not prospectively divide patients into radiation absorbed dose categories of > or <100Gy; this was a determined based on post-infusion dosimetry measurements, as described in the Treatment section (Method) and then correlated with overall survival, among others. Regarding the 8 months standard of care following recurrence, the Introduction currently states: *Median overall survival (mOS) with first progression is typically ~7.4-9.2 months^{7,8} and with second progression, approximately four months⁹.*

The following references are used as support:

- Friedman, H. S. *et al.* Bevacizumab Alone and in Combination with Irinotecan in Recurrent Glioblastoma. *JCO* 27, 4733–4740 (2009).
- Cloughesy, T. F. *et al.* A randomized controlled phase III study of VB-111 combined with bevacizumab vs bevacizumab monotherapy in patients with recurrent glioblastoma (GLOBE). *Neuro-Oncology* 22, 705–717 (2020).
- Quant, E. C. *et al.* Role of a second chemotherapy in recurrent malignant glioma patients who progress on bevacizumab. *Neuro-Oncology* 11, 550–555 (2009).

3. The choice of 186Re (with a half-life of ~89.2 hours) is made here. It is a reactor-produced isotope that can also be generated by cyclotron. Can the authors comment on the interest, advantages and disadvantages of this product. On this subject, the constraint for clinicians and patients (37 days for zero radioactivity) is not mentioned? Could the same type of work have been done with 188Re which is a generator product and has a half-life of ~16.9 hours (total decay in 7 days)?

We have revised the text in the introduction to state: ¹⁸⁶Re was preferentially chosen over ¹⁸⁸Re, which is generator produced, due to its favorable mean path length of 1.8 mm (compared to ¹⁸⁸Re's 3.5 mm) to better avoid healthy tissue penetration for lower normal tissue toxicity²². Additionally, the authors previously reported on ¹⁸⁶RNL preclinically²⁰.

The following reference was used as support:

- Uccelli, L. *et al.* Rhenium Radioisotopes for Medicine, a Focus on Production and Applications. *Molecules* 27, 5283 (2022).
- Phillips, W. T. *et al.* Rhenium-186 liposomes as convection-enhanced nanoparticle brachytherapy for treatment of glioblastoma. *Neuro-Oncology* 14, 416–425 (2012).

4. In link with the above, it is essential to discuss the choice of the radiopharmaceutical (¹⁸⁶Re-nanoliposome). In this regard, it is important to highlight recent preclinical developments more broadly on the keywords: radiotherapy, rhenium, nanoparticle, glioma (eg. DOI: 10.7150/thno.19403; DOI: 10.1016/j.biomaterials.2011.05.067).

We have revised the introduction to incorporate the key publications that informed the authors prior to study start, now stating: Prior to study start we searched PubMed from any time up to December 1, 2013 for clinical therapeutic studies using ¹⁸⁶Re to treat glioma using the search terms “(¹⁸⁶Re[ALL FIELDS] AND glioma[ALL FIELDS]) and (rhenium-186[ALL FIELDS] AND glioma[ALL FIELDS]). The search gave four publications^{23–26}. Intramedullary cystic spinal cord pilocytic astrocytoma was managed with minor side effects using ¹⁸⁶Re intracavitary irradiation, with stabilization of the cyst and neurological deficit improvement²³. Combination treatment for pilocytic astrocytoma using ¹⁸⁶Re delivered 400 Gy to the cyst wall and resulted in progressive cyst disappearance and mural nodule retraction²⁶. A fibrin glue of ¹⁸⁸Re and ¹⁸⁶Re bound in microspheres was used post-tumor resection in a 9L-glioblastoma rat model. 60% of treated animals survived 36 days, compared to control animals (17 ± 3 days)²⁴. Lastly, intracavitary ¹⁸⁶Re application was used in six cases of cystic craniopharyngiomas but abandoned because of cyst recurrence and leakage²⁵.

The following references were used as support:

- Colnat-Coulbois, S., Klein, O., Braun, M., Thouvenot, P. & Marchal, J.-C. Management of Intramedullary Cystic Pilocytic Astrocytoma with Rhenium-186 Intracavitary Irradiation: Case Report. *Neurosurgery* 66, E1023–E1024 (2010).
- Häfeli, U. O., Pauer, G. J., Unnithan, J. & Prayson, R. A. Fibrin glue system for adjuvant brachytherapy of brain tumors with ¹⁸⁸Re and ¹⁸⁶Re-labeled microspheres. *Eur J Pharm Biopharm* 65, 282–288 (2007).
- Gahbauer, H. *et al.* Combined use of stereotaxic CT and angiography for brain biopsies and stereotaxic irradiation. *AJNR Am J Neuroradiol* 4, 715–718 (1983).
- Proust, F. *et al.* [Combination treatment for pilocytic astrocytoma: stereotaxic radiosurgery and endocavitary radiotherapy]. *Neurochirurgie* 44, 50–54 (1998).

5. The radiation absorbed dose calculation uses the MIRDose algorithm which assumes 100% of the beta-radiation dose has deposited locoregionally. It is not perfectly clear how the dose distribution study attest to this point (volume of ¹⁸⁶RNL distributed in the brain and in the tumor, percentage of tumor volume treated, radiation absorbed dose to the tumor, and whole-body normal organ doses)?

The MIRDose algorithm has been used as a standard for clinically calculating radiation absorbed doses in normal organs. The corresponding OLINDA software used is an FDA-approved software for normal organ radiation absorbed dose calculation. The assumption is an embedded component of the MIRDose algorithm. We therefore consider that the assumption used by MIRDose is based on the following: 1) there is only < 1mm-5mm range in the tissue with beta-particle therapeutic radiation depending on the energies of the beta-particle (there is only 1.8 mm average range for the beta-radiation from ¹⁸⁶Re); 2) the nuclear imagers for clinical patients has poorer image resolution (~ 8 mm to 1 cm) than the radiation range; and 3) for the evaluation of normal organ effect, more micro-scale radiation dose calculation may not be necessary, if the normal organ doses are not high. Meanwhile, we understood the evaluation of radiation absorbed dose distribution, especially the coverage of radiation therapy, can be very important. Accordingly, the specific methodology for the evaluation of locoregional distribution of therapeutic agent at the tumor region has been developed. The data have been evaluated to investigate its relationship with therapy effect, like patient survival, which has also been described in the manuscript.

6. Regarding, MGMT status, it has indeed been demonstrated that it impacts the effectiveness of temozolomide (TMZ), which however was not used in this study. Does the author have any evidence that it can define 186Re-radioresponse which will give echoes of how the data produced here on this subject could be further used (larger cohorts) in terms of prognosis feature.

We thank the reviewer for the insightful question. We have requested archival specimens which can be analyzed retrospectively to identify prognostic factors. DNA damage response pathways are of keen interest, especially the ATR-CHK1 pathway, due to stabilizing replication forks encountering oxidatively-damaged DNA or DSBs. Inhibiting replication fork arrest and then restart after repair would lead to synergy with the Rhenium Obisbameda. At present, the number of data points is low which precludes meaningful discovery. As we acquire additional survival data and archival specimens in the Phase 2 trial, we plan to analyze this further. Additionally, in an ongoing study of rhenium obisbameda for leptomeningeal metastases we are able to collect tumor cells at various time points after treatment. This is allowing us to profile response and identify DDR alterations that may correlate with response. We look forward to presenting those findings at a later date.

7. Although this reviewer is not a statistician, it seems important to include the contribution of an expert in the field in this study. For example, there is no mention of calculated sample size, hazard ratio (HR) or randomization. Can you clarify this point?

A statistician contributed to the design of the study and its analysis. We have revised the text in the Statistics section (Methods) to state: *Discrete outcomes were summarized with frequencies and percentages, and continuously distributed outcomes with the mean and standard deviation (SD). Absorbed dose was dichotomized to “ ≥ 100 Gy and < 100 Gy” as defined by absorbed dose ≥ 100 and absorbed dose < 100 , respectively. The significance of variation in the mean and cohort was assessed with analysis of variance. The significance of associations between categorical outcome and cohort was assessed with chi-square and Fisher’s exact tests. Variation in survival with category of absorbed dose was assessed, without covariate adjustment, with log rank tests and with accelerated failure time (AFT) models with adjustment for covariates; a lognormal error term was assumed. All measurements were made on individual patients. No repeated measures within patient are shown. Covariates were used in the AFT models and are specified in the summary tables. Except for the AFT models, where the logarithm of survival (PFS and OS) is modelled in terms of covariates, all statistical analyses were done in original units. Measures of central tendency such as the mean and median are specified in the tables. Fold change was defined as the antilog of the beta coefficient in the AFT model. No Bayesian methods and no hierarchical models were used. The sample sizes were specified in accordance with 3+3 methodology; power calculations were not made and were not used. Corrections for multiple comparisons were not applied. All significance testing was 2-sided with a significance level of 5%. SAS Version 9.4 (SAS Institute, Cary NC) was used.*

Reviewer #4

We thank Reviewer #4 for their thoughtful consideration of our manuscript and provide the following supporting information to address their open questions.

The authors reported the results of the Phase I dose escalation trial NCT01906385 that evaluated a single dose of 186RNL administered through a convection enhanced delivery catheter in participants with recurrent Glioma (GBM). The study employed a modified 3+3 Fibonacci dose escalation method. The maximum tolerated dose (MTD) was not reached, and overall survival was found to correlate with both the percentage of tumor volume coverage by the drug and the radiation absorbed dose to the tumor. The manuscript is well-written. We suggest adding a few more details on the study design and results for rigor and reproducibility, and for interpretation of the results, as recommended by the Dose Finding CONSORT extension (DEFINE-CONSORT; BMJ 2023 Oct 20;383:e076387. doi: 10.1136/bmj-2023-076387.)

1. The definitions of the endpoints should be included and/or clarified:
 - a. The DLT definition, even if it follows the standard definition based on grade 3 of the CTCAE scale, should be clearly stated, including the DLT evaluation window.

We have revised the text in the new Safety section to state: *For this study, a dose limiting toxicity (DLT) was defined as grade 3 or greater acute CNS toxicity attributable to the study intervention which persists for 96 hours or more (see below discussion of delayed events) OR grade 3 or greater non-CNS toxicity which is attributable to the study*

intervention, as per the grading scale of the grading scale of the CTCAE v4.0²⁹. The standard DLT window was 28 days following treatment. Additionally, given the possibility for radiation effects outside of the standard 28-day DLT window, additional consideration was given that extended the DLT evaluation period for CNS toxicity to 90 days between successive cohorts. Lastly, if a patient within a cohort experienced a CNS toxicity that would be defined as dose limiting, the entire cohort would complete 90 days evaluation before the successive cohort would commence.

The following reference was used in support:

- National Institute of Cancer (NCI). Common Terminology Criteria for Adverse Events (CTCAE), v4.0. <http://evs.nci.nih.gov/ftp1/CTCAE/About.html>.

b. The time point for the absorbed dose could be stated more clearly.

We currently note in the Treatment section the following: *Finally, cumulated radioactivity (\bar{A}) in the tumor within 192 hours of infusion was calculated and used to determine the mean radiation absorbed dose in the tumor.* The radiation absorbed dose is a cumulation of data over a time period (start time through 192 hours), and not taken from a specific time point.

2. The timepoint at which the volume of distribution was obtained can also be clarified.

As noted above, the volume of distribution is determined by using a cumulation of data over 192 hours post drug infusion.

3. While the paper mentions that the 3+3 design is used, it does not clearly state the dose-escalation decisions and the definition of MTD. For example,

a. Whether other endpoints were considered for the dose escalation (e.g., pharmacokinetic measures).

The primary objective of the study was to determine the maximum tolerated dose of ¹⁸⁶RNL by convection enhanced delivery (CED) at the time of planned stereotactic biopsy, when necessary, as standard of care. The secondary objectives were to assess the safety of single dose ¹⁸⁶RNL by CED; assess the dose distribution of ¹⁸⁶RNL by CED; determine the overall response rate by Radiographic Assessment in Neuro-Oncology (RANO) criteria following ¹⁸⁶RNL treatment; determine disease specific progression-free survival after ¹⁸⁶RNL treatment; and determine overall survival (OS) after ¹⁸⁶RNL treatment. This is also noted in the Introduction section, where it states: *Following these studies, we initiated ReSPECT-GBM, a multicenter, sequential cohort, open-label, volume and dose escalation study of the safety, tolerability, and distribution of a single dose of ¹⁸⁶RNL given by CED for recurrent glioma. The primary objective was to determine the maximum tolerated dose (MTD) of ¹⁸⁶RNL by CED in patients with recurrent glioma. Secondary objectives included the assessment of the safety and tolerability of a single dose ¹⁸⁶RNL, the dose distribution of ¹⁸⁶RNL by imaging, the overall response rate (ORR), disease-specific progression-free survival (PFS), and OS. A graphical summary of the trial design and its outcomes is included in Figure S4.*

b. The 3+3 would not have 6 patients in cohort 6 in the absence of DLT. What was the rationale for adding three more patients in cohort 6 (with increasing maximum flow rate)? Was this pre-specified in the protocol?

We have revised the text in the Discussion section to state: *Following interim analysis of cohort 6, we observed a plateau from cohort 5 to cohort 6 in both dose distribution and absorbed dose. As a result of this finding, the recommendation of the Data Safety Monitoring Board (DSMB) was to expand cohort 6 to confirm tolerability of 22.3mCi administered dose in 8.8mL infusate volume as the Phase 2 recommended dose (RP2D) to target both bulk disease and adjacent microscopic disease in patients with one recurrence and tumor sizes of 20 cm³ or less. Once completed, and without the MTD being reached, all cohort data was analyzed with recommendation from the DSMB to proceed with Phase 2 utilizing this dose.* This was done under a protocol amendment and not pre-specified.

4. The dose levels to be evaluated should be specified in the methods along with the rationale for them and the starting dose. It is not clear if:

a. The Fibonacci was used on RNL activity and the infused volume was calculated from that.

The modified Fibonacci was based on the administered dose. We have revised the text in the Demographics section (Results) to state: *The study used a modified 3+3 dose escalation²⁷, with increased in total radioactivity by doubling from 1 mCi in Cohort 1 to 8 mCi in Cohort 4, followed by a 66% increase in cohorts 5 and 6. The administered dose range was 1 mCi in a volume of 0.66 mL through 22.3 mCi in 8.80 mL (Table 1).* It should be emphasized that the safety of each cohort was review by an independent Data Safet Monitoring Board who made recommendations on the dose escalation. Given the lack of significant AEs in the first 4 cohorts, the DSMB, dose doubling was recommended.

The following reference was used in support:

- Le Tourneau, C., Lee, J. J. & Siu, L. L. Dose Escalation Methods in Phase I Cancer Clinical Trials. *JNCI: Journal of the National Cancer Institute* 101, 708–720 (2009).

b. The methods should also clarify the relationship between the assigned and absorbed dose.

We have tried to clarify the difference between the administered dose and the radiation absorbed dose throughout the manuscript.

5. Whether a sentinel patient was included or not.

In the context of a sentinel patient as a marker of toxicity, we did not. As mentioned, the toxicity was modest and as such there was not specific toxicity to follow in other patients.

6. We appreciated the results presented both by cohort and by the total number of trial patients. In general, the results are well-reported and discussed. Further clarifications on:

a. The number of catheters used by cohort and in total should be reported as this was left to the discretion of the treating physician and it is not clear if it differed by cohort.

We have revised the text in the Treatment section (Methods) to state: *Once patients were consented, they underwent treatment planning MRI within seven days before drug infusion to evaluate tumor characteristics, including location, structure, shape, and dimension. Based on the treatment planning MRI, patients were further selected for study status. If the tumor had progressed beyond the screening criteria, they were considered a screen failure. Once confirmed to be appropriate for CED, stereotactic treatment planning was centralized and performed by the Principal Investigator (PI) together with the local neurosurgeon. The tumor volume was calculated by the use of iPlan Flow, as was all in silico treatment planning and convection simulation, as has been previously described¹⁸. Briefly, using the iPlan software, the PI would determine the number of catheters and their trajectories, based on the size and location of the tumor, in accordance with pre-set parameters noted below. Early cohorts minimized the number of catheters for safety; once this was determined, the number of catheters used (1-4) was at the treating investigator's discretion based on planning to allow for suitable coverage of the enhancing tumor with surrounding T2 FLAIR abnormality. Immediately following a SoC stereotactic biopsy procedure for confirmation of disease progression, BrainLab Flexible CED catheter(s) (7.5 mm tip) were placed, with at least one catheter proceeding along the same needle track as the SoC biopsy. Catheter placement was optimized to avoid ependymal surfaces by >0.5 cm as they offer no resistance to fluid flow and 2.0 cm from the closest sulcus, fissure, resection cavity, or cortical surface. Distal placement of the infusing tip within the tumor was also critical to avoid the pressure gradient from the tumor core to periphery to adjacent brain. Postoperative head CT was performed to evaluate for hematoma or pneumocephalus.*

To allow for fibrin and clot deposition to occur, ¹⁸⁶RNL infusion was performed approximately 24 hours following catheter placement. ¹⁸⁶RNL was manufactured as previously described²⁰. Super-saturated potassium iodide (SSKI, 600 mg) was administered by mouth with water or juice prior to infusion. The infusion rate started at 1 µl/min and stepped to 20 µl/min (Table S6) using a syringe pump (Medfusion 3500 and 4000, Adepto Medical, Kansas City, MO). The total time of infusion was a function of flow rate, number of catheters used, and total volume infused (Table S6, Table S7, Table 1). During infusion, planar and tomographic images were collected using a dual-detector SPECT/CT camera. A sealed vial with known ¹⁸⁶RNL was positioned next to the patient at each time of image acquisition for radioactivity quantification. Dynamic images were acquired via real-time persistence scope for evaluation of focal accumulation of activity at the assumed tip of the catheter(s). When activity was observed to accumulate focally, that time was designated as the beginning of the planned therapeutic volume infusion to correct for dead space in the catheter line. Catheters deemed

unsatisfactory (backflow along the catheter or spillage into adjacent CSF space) were stopped and the remaining volume switched to the remaining catheters.

To minimize backflow concerns³¹, we utilized the Brainlab catheter, which is both flexible and has a “step design” at the tip, avoided resection cavities and pial surfaces, and employed a slow ramping of infusion rates (Table S6). Furthermore, backflow was monitored during infusion using planar imaging as mentioned above, allowing changes to be made in real-time to ensure the planned administered dose was delivered in full.

The following reference was used as support:

- Wembacher-Schroeder, E. *et al.* Evaluation of a patient-specific algorithm for predicting distribution for convection-enhanced drug delivery into the brainstem of patients with diffuse intrinsic pontine glioma. *J Neurosurg Pediatr* **28**, 34–42 (2021).

- b. Report the AE by cohort using patient as unit of analysis to know what proportion of patients experienced those grades of AE in each cohort. The frequency of events using events are unit (Tables 4 and 5) can be supplemental.

We have revised the AE tables accordingly.

- c. Summary statistics on the absorbed doses per cohort.

We have provided a table for absorbed doses per patient/cohort (Table S8) and a figure of the summary per cohort (Figure 1).

- d. In Figure 1 (page 6), while the median Vd in cohort 6 is higher compared to all other cohorts, the maximum Vd is observed in a patient in cohort 5. Do the authors have an explanation for this?

We have revised the text in the Discussion section to state: *Following interim analysis of cohort 6, we observed a plateau from cohort 5 to cohort 6 in both dose distribution and absorbed dose. As a result of this finding, the recommendation of the Data Safety Monitoring Board (DSMB) was to expand cohort 6 to confirm tolerability of 22.3mCi administered dose in 8.8mL infusate volume as the Phase 2 recommended dose (RP2D) to target both bulk disease and adjacent microscopic disease in patients with one recurrence and tumor sizes of 20 cm³ or less. Once completed, and without the MTD being reached, all cohort data was analyzed with recommendation from the DSMB to proceed with Phase 2 utilizing this dose.*

- e. For survival analysis, was the dichotomization (<100 Gy or not) decided before the study onset, or is it a post-hoc analysis? Why was the survival analysis not done by assigned dose/cohort like ORR, but by absorbed dose?

We have tried to clarify the difference between the dose administered and radiation absorbed dose throughout the manuscript. While the study prospectively assigned patients to cohorts differing in administered dose, the study did not prospectively divide patients into radiation absorbed dose categories of > or <100Gy. It was not possible to know *a priori* what absorbed dose would be for a given administered dose as this is highly dependent on both distribution and retention. Indeed, this was a determined on a case by case basis using post-infusion dosimetry measurements, as described in the Treatment section and then correlated with overall survival, among others. Survival analysis by Cohort – or administered dose – did not show any correlation with overall survival, as the enhancing tumor volume was required to be within the treatment field volume for a respective cohort, per the inclusion criteria. We therefore presented the survival data as we did, for all patients and the dichotomization, which we saw in our preclinical studies. We have revised the Results section to state: *Given finding from our preclinical studies identifying 100Gy as a threshold for survival benefit²⁰, patients were dichotomized by absorbed dose of 100 Gy.* Likewise, in the Introduction section, we state: *Furthermore, >100 Gy of ¹⁸⁶RNL was shown to eradicate grafted tumors and prolong overall survival, without clinical or microscopic evidence of toxicity²⁰.*

The following reference was used as support:

- Phillips, W. T. *et al.* Rhenium-186 liposomes as convection-enhanced nanoparticle brachytherapy for treatment of glioblastoma. *Neuro-Oncology* **14**, 416–425 (2012).

- f. For Table S4, did the authors examine the correlation between “Ratio of Treated to Total Tumor Volume”, “Total Dose in Distribution Volume”, and “Volume Administered” for the “Progression-Free Survival Accelerated Failure Time Model”? It is not clear why both of this need to be in the model.

We thank the reviewer for the question, but we are not completely sure of the intended meaning of the question. We believe this question to mean that some of the variables assessed are redundant. The first variable was to assess coverage independent of the dose administered, the second to assess the absorbed dose, and the third an ITT based on what was given rather than achieved.

- g. The trial accrued more males than females and mostly white patients; this could be discussed in terms of generalizability.

We thank the reviewer and note that the study population was reflective of the patient population in the centers participating. We have revised the text in a new Sex and Gender section to address the information regarding male/female accrual to state: *Sex and gender, as determined based on self-reporting, was not a factor in patient inclusion criteria, nor were there sufficient numbers of patients of either gender to make a meaningful sex- and gender-based conclusions in the study; as such, sex- and gender-based analyses have not been reported here.* Based on the small number of patients, we also extend this same rationale to race/ethnicity.

Nature Communication Manuscript Response

Title: Convection Enhanced Delivery of Rhenium (¹⁸⁶Re) Obisbameda in Recurrent Glioma: a multicenter, single arm, phase 1 clinical trial

Corresponding Author: Andrew Brenner, MD, PhD

Manuscript Revision: NCOMMS-24-25017-C

Reviewer #3

We thank Reviewer #3 for their thoughtful consideration of our revised manuscript and our response.

1. The manuscript deserves publication in Nature Communications considering the improvements made by the authors.

Reviewer #4

We thank Reviewer #4 for their thoughtful consideration of our revised manuscript and our response and provide the following supporting information to address their open item.

1. While the most of the new revisions have addressed my previous comments, I think it is important to address question 4 and note in the results and discussion that survival outcomes were not associated with administered dose. This is very important and worthy to be discussed as it is relevant to the reproducibility of a dose-finding clinical trial. [Question 4 from Response 2: It is important to note that survival was not associated with administered dose and only with absorbed dose in the paper. This is very important and should be clearly explained and discussed. Absorbed dose is an outcome which could be dependent on many factors.

We have added the following text to the discussion to address the Reviewer’s comment: “...a correlation between administered dose and survival was not observed, and likely reflects the decision to match tumor volume in the inclusion criteria to the volume of administration while holding the concentration constant through the first four cohorts.”

Original Response 2 for Reviewer 4 for Reference

Reviewer #4

We thank Reviewer #4 for their thoughtful consideration of our revised manuscript and our response and provide the following supporting information to address their open questions.

2. Based on the revision, “Given the lack of significant AEs in the first 4 cohorts, the DSMB, dose doubling was recommended.” The manuscript should state that cohorts 5 and 6 were added later after review of data from first 4 cohorts as doses in phase 1 trials are generally pre-specified for rigor and reproducibility. Please specify the set of doses that were specified at the start of the trial in the treatment section and those that were added after review and the rationale for those dose selection.

We apologize for the confusion caused by our prior response. The dose escalation scheme was determined prior to study start; however, the decision to escalate to the next Cohort was dependent on the determination of the DSMB’s review of any given Cohort. We did not deviate from the initial plan.

Cohort	Administered Dose (mCi)	Percent Change
1	1.0	NA
2	2.0	100%
3	4.0	100%
4	8.0	100%
5	13.4	67%
6	22.3	67%

We have revised the text in the Results, Demographics section to state: *The study used a modified 3+3 dose escalation³¹, with increased in total radioactivity by doubling from 1 mCi in Cohort 1 to 8 mCi in Cohort 4, followed by a 67% increase in Cohorts 5 and 6. All administered doses were determined prior to study start, with dose escalation to a subsequent Cohort following confirmation by the Data Safety Monitoring Board (DSMB), after a review of all safety data. The administered dose range was 1 mCi in a volume of 0.66 mL through 22.3 mCi in 8.80 mL (Table 1).*

The following reference was used as support:

- Le Tourneau, C., Lee, J. J. & Siu, L. L. Dose Escalation Methods in Phase I Cancer Clinical Trials. *JNCI: Journal of the National Cancer Institute* 101, 708–720 (2009).

3. Table 2 for the AE is still by events not patients. The unit of analysis is still events to have 24 mild events in the first cohort with 3 patients. Please specify the number of AE out of the 3 patients taking the maximum grade of AE.

We have updated all AE tables accordingly with number per Cohort specified.

4. The variability in the absorbed dose is of concern (Table S8) and not appropriate to summarize at mean +/- SEM in Figure 1. Given the small sample size and large variability per cohort it is best to graph all points. Please explain the poor correlation between administered dose and absorbed dose. Was this evaluated? How reproducible is the absorbed dose?
5. It is important to note that survival was not associated with administered dose and only with absorbed dose in the paper. This is very important and should be clearly explained and discussed. Absorbed dose is an outcome which could be dependent on many factors.

For Questions 3 and 4: We apologize for any confusion in the manuscript regarding administered dose and absorbed dose. Administered dose (mCi) was reproducibly manufactured and quality controlled as outlined in the Chemistry, Manufacturing, and Controls (CMC) section of the IND (and described in brief in Methods, ¹⁸⁶RNL Manufacturing) to ensure that the radioactivity delivered to the patient was standardized per Cohort, i.e., 6.6 mCi was delivered to each patient in Cohort 1.

There were some clear factors that would result in non-proportional increase in absorbed dose with administered dose. First, due to FDA clinical reviewers concerns about high doses to larger tumors, a proportional increase in volume was included in the dose escalation with activity leading to a stable concentration administered through Cohort 4. This means that as we increase dose, the distribution increased proportionately. The volume was held steady at Cohort 5, which resulted in a proportionate increase in absorbed dose. Further, early Cohorts (particularly Cohorts 2 and 3) included patients with progression on bevacizumab. As discussed in the manuscript, bevacizumab impacted CED and resulted in poor absorbed doses, leading to it being added as an exclusion criterion. Finally, as we gained experience with the safety and distribution, additional catheters were used to aid in distribution that adds additional variability. We captured this information in the Methods, Treatment section:

The tumor volume was calculated by the use of iPlan Flow, as was all in silico treatment planning and convection simulation, as has been previously described¹⁸. Briefly, using the iPlan software, the PI would determine the number of catheters and their trajectories, based on the size and location of the tumor, in accordance with pre-set parameters noted below. Early cohorts minimized the number of catheters for safety and to characterize the distribution; once this was determined, the number of catheters used (1-4) was based on planning to allow for suitable coverage of the enhancing tumor with surrounding T2 FLAIR abnormality. Immediately following a SoC stereotactic biopsy procedure for confirmation of disease progression, BrainLab Flexible CED catheter(s) (7.5 mm tip) were placed, with at least one catheter proceeding along the same needle track as the SoC biopsy. Catheter placement was optimized to avoid ependymal surfaces by >0.5 cm as they offer no resistance to fluid flow and 2.0 cm from the closest sulcus, fissure, resection cavity, or cortical surface. Distal placement of the infusing tip within the tumor was also critical to avoid the pressure gradient from the tumor core to periphery to adjacent brain.

Absorbed dose to the tumor per Cohort, therefore, was a function of a number of factors outside the administered dose, which was constant per Cohort. In general, we noted that where pre-treatment planning could be optimally maximized to deliver the administered dose to the tumor and only the tumor the absorbed dose to the tumor was high. In instances where the absorbed dose to the tumor was lower than another patient in the same Cohort, it could be a function of drug diffusing

away to an area of low resistance like a nearby resection cavity, non-optimal catheter placement due to the safety planning rules in the protocol, or a tumor that was simply more maximally covered by the volume of distribution because it was smaller than another tumor. In other words, personalized pre-treatment planning is a critical component of Rhenium (^{186}Re) Obisbameda for the treatment of recurrent glioblastoma.

Table S8 was provided to provide the intra-cohort variability – based on what is noted above – and Figure 1 describes the relationship between the volume of distribution and the absorbed dose.

Lastly, we attempted to describe these issues in the Discussion section:

In this study the maximum absorbed dose was ~740 Gy. Both preclinical dose finding studies and statistical modeling of the clinical data with long rank testing suggested a dose effect of greater than 100Gy. When this dose threshold was applied, a significant marginal correlation was observed between dose, both in its binary (<100 Gy, >100 Gy) and continuous form, and OS. A similar marginal correlation was also observed between percent tumor coverage and OS. Furthermore, these correlations remained statistically significant after adjustment for baseline factors in multivariate models. While further assessment of the data using a combination of perfusion and delayed contrast imaging is ongoing, we have selected a primary endpoint of OS for our Phase 2 study, currently underway.

Although we found a correlation between the number of catheters, volume of distribution, and the administered volume, we did not observe a correlation between number of catheters and percent tumor coverage. There are a number of confounders that may impact correlation between number of catheters and percent tumor coverage, particularly that treatment planning for larger tumors likely would drive a decision to place a higher number of catheters, but that the inherent challenges of larger tumors would make coverage less successful. Coverage of greater than 70% appeared to be associated with benefit (Figure S1b), but this is likely a factor of the fixed thresholding method used relative to maximum voxel radioactivity (count) of each SPECT image, and that treatment effects extend beyond this region which may be better discriminated with 3D dosimetry.

Several limitations were present in this study which will require further evaluation. First, since this is a first-in-human dose escalation study, including varying doses, administration rates, number of catheters, and tumor sizes, any results should be taken as preliminary and will require further validation. From a dose distribution perspective, SPECT as an imaging modality to be used for dosimetry and absorbed dose quantification has inherent technological limitations when addressing CNS tumors based on camera resolution, voxel size, and brain anatomy. Further, distribution was estimated based on activity rather than isodose lines. We have since begun performing three-dimensional dosimetry which can define treatment volumes based upon absorbed dose isodose lines and will better define biologically relevant distribution. CED also is technically challenging and can be a source of treatment failure even with an efficacious therapeutic. We recognize the infrastructure required, including sophisticated, cost-intensive neuronavigation software and hardware. At present, this technology from companies such as Brainlab or Stryker are currently available in most hospitals with neuro-oncology expertise, and routinely used. For this study and our ongoing Phase 2 trial, treatment planning is performed centrally with participation of the local neurosurgeon. As reported, following early cohorts, multiple catheters were used whenever possible to maximize distribution, minimize convection time, improve coverage of irregular shaped tumors, and to provide redundancy in case of catheter failure. The ability to visualize the delivery of the therapeutic directly at the time of administration is a unique advantage of CED with a radiopharmaceutical, and further analysis of the entire data set is ongoing and will be reported. Finally, determining response can be challenging in CNS tumors, particularly with radiation which can result in pseudo-progression. To better characterize response, we are performing advanced imaging modalities including Delta T1 subtraction mapping, DSC perfusion, and delayed contrast mapping (TRAMs). In preliminary parametric analyses we observed a statistically significant difference in repeated pair measurements with untreated tumor volume significantly increased relative to treated tumor volume ($p < 0.0001$) suggesting that progression arises outside the treated volume.

Original Response 1 for Reviewer 4 for Reference

Reviewer #4

We thank Reviewer #4 for their thoughtful consideration of our manuscript and provide the following supporting information to address their open questions.

The authors reported the results of the Phase I dose escalation trial NCT01906385 that evaluated a single dose of ^{186}RnL administered through a convection enhanced delivery catheter in participants with recurrent Glioma (GBM). The study employed a modified 3+3 Fibonacci dose escalation method. The maximum tolerated dose (MTD) was not reached, and overall survival was found to correlate with both the percentage of tumor volume coverage by the drug and the radiation absorbed dose to the tumor. The manuscript is well-written. We suggest adding a few more details on the study design and results for rigor and reproducibility, and for interpretation of the results, as recommended by the Dose Finding CONSORT extension (DEFINE-CONSORT; BMJ 2023 Oct 20;383:e076387. doi: 10.1136/bmj-2023-076387.)

1. The definitions of the endpoints should be included and/or clarified:

- a. The DLT definition, even if it follows the standard definition based on grade 3 of the CTCAE scale, should be clearly stated, including the DLT evaluation window.

We have revised the text in the new Safety section to state: *For this study, a dose limiting toxicity (DLT) was defined as grade 3 or greater acute CNS toxicity attributable to the study intervention which persists for 96 hours or more (see below discussion of delayed events) OR grade 3 or greater non-CNS toxicity which is attributable to the study intervention, as per the grading scale of the CTCAE v4.0²⁹. The standard DLT window was 28 days following treatment. Additionally, given the possibility for radiation effects outside of the standard 28-day DLT window, additional consideration was given that extended the DLT evaluation period for CNS toxicity to 90 days between successive cohorts. Lastly, if a patient within a cohort experienced a CNS toxicity that would be defined as dose limiting, the entire cohort would complete 90 days evaluation before the successive cohort would commence.*

The following reference was used in support:

- National Institute of Cancer (NCI). Common Terminology Criteria for Adverse Events (CTCAE), v4.0. <http://evs.nci.nih.gov/ftp1/CTCAE/About.html>.

- b. The time point for the absorbed dose could be stated more clearly.

We currently note in the Treatment section the following: *Finally, cumulated radioactivity (\bar{A}) in the tumor within 192 hours of infusion was calculated and used to determine the mean radiation absorbed dose in the tumor.* The radiation absorbed dose is a cumulation of data over a time period (start time through 192 hours), and not taken from a specific time point.

2. The timepoint at which the volume of distribution was obtained can also be clarified.

As noted above, the volume of distribution is determined by using a cumulation of data over 192 hours post drug infusion.

3. While the paper mentions that the 3+3 design is used, it does not clearly state the dose-escalation decisions and the definition of MTD. For example,

- a. Whether other endpoints were considered for the dose escalation (e.g., pharmacokinetic measures).

The primary objective of the study was to determine the maximum tolerated dose of ^{186}RnL by convection enhanced delivery (CED) at the time of planned stereotactic biopsy, when necessary, as standard of care. The secondary objectives were to assess the safety of single dose ^{186}RnL by CED; assess the dose distribution of ^{186}RnL by CED; determine the overall response rate by Radiographic Assessment in Neuro-Oncology (RANO) criteria following ^{186}RnL treatment; determine disease specific progression-free survival after ^{186}RnL treatment; and determine overall survival (OS) after ^{186}RnL treatment. This is also noted in the Introduction section, where it states: *Following these studies, we initiated ReSPECT-GBM, a multicenter, sequential cohort, open-label, volume and dose escalation study of the safety, tolerability, and distribution of a single dose of ^{186}RnL given by CED for recurrent glioma. The primary objective was to determine the maximum tolerated dose (MTD) of ^{186}RnL by CED in patients with recurrent glioma. Secondary objectives included the assessment of the safety and tolerability of a single dose ^{186}RnL , the dose distribution of ^{186}RnL by imaging,*

the overall response rate (ORR), disease-specific progression-free survival (PFS), and OS. A graphical summary of the trial design and its outcomes is included in Figure S4.

- b. The 3+3 would not have 6 patients in cohort 6 in the absence of DLT. What was the rationale for adding three more patients in cohort 6 (with increasing maximum flow rate)? Was this pre-specified in the protocol?

We have revised the text in the Discussion section to state: *Following interim analysis of cohort 6, we observed a plateau from cohort 5 to cohort 6 in both dose distribution and absorbed dose. As a result of this finding, the recommendation of the Data Safety Monitoring Board (DSMB) was to expand cohort 6 to confirm tolerability of 22.3mCi administered dose in 8.8mL infusate volume as the Phase 2 recommended dose (RP2D) to target both bulk disease and adjacent microscopic disease in patients with one recurrence and tumor sizes of 20 cm³ or less. Once completed, and without the MTD being reached, all cohort data was analyzed with recommendation from the DSMB to proceed with Phase 2 utilizing this dose. This was done under a protocol amendment and not pre-specified.*

4. The dose levels to be evaluated should be specified in the methods along with the rationale for them and the starting dose. It is not clear if:

- a. The Fibonacci was used on RNL activity and the infused volume was calculated from that.

The modified Fibonacci was based on the administered dose. We have revised the text in the Demographics section (Results) to state: *The study used a modified 3+3 dose escalation²⁷, with increased in total radioactivity by doubling from 1 mCi in Cohort 1 to 8 mCi in Cohort 4, followed by a 66% increase in cohorts 5 and 6. The administered dose range was 1 mCi in a volume of 0.66 mL through 22.3 mCi in 8.80 mL (Table 1).* It should be emphasized that the safety of each cohort was reviewed by an independent Data Safety Monitoring Board who made recommendations on the dose escalation. Given the lack of significant AEs in the first 4 cohorts, the DSMB, dose doubling was recommended.

The following reference was used in support:

- Le Tourneau, C., Lee, J. J. & Siu, L. L. Dose Escalation Methods in Phase I Cancer Clinical Trials. *JNCI: Journal of the National Cancer Institute* 101, 708–720 (2009).

- b. The methods should also clarify the relationship between the assigned and absorbed dose.

We have tried to clarify the difference between the administered dose and the radiation absorbed dose throughout the manuscript.

5. Whether a sentinel patient was included or not.

In the context of a sentinel patient as a marker of toxicity, we did not. As mentioned, the toxicity was modest and as such there was not specific toxicity to follow in other patients.

6. We appreciated the results presented both by cohort and by the total number of trial patients. In general, the results are well-reported and discussed. Further clarifications on:

- a. The number of catheters used by cohort and in total should be reported as this was left to the discretion of the treating physician and it is not clear if it differed by cohort.

We have revised the text in the Treatment section (Methods) to state: *Once patients were consented, they underwent treatment planning MRI within seven days before drug infusion to evaluate tumor characteristics, including location, structure, shape, and dimension. Based on the treatment planning MRI, patients were further selected for study status. If the tumor had progressed beyond the screening criteria, they were considered a screen failure. Once confirmed to be appropriate for CED, stereotactic treatment planning was centralized and performed by the Principal Investigator (PI) together with the local neurosurgeon. The tumor volume was calculated by the use of iPlan Flow, as was all in silico treatment planning and convection simulation, as has been previously described¹⁸. Briefly, using the iPlan software, the PI would determine the number of catheters and their trajectories, based on the size and location of the tumor, in accordance with pre-set parameters noted below. Early cohorts minimized the number of catheters for safety; once this was determined, the number of catheters used (1-4) was at the treating investigator's discretion based on planning to*

allow for suitable coverage of the enhancing tumor with surrounding T2 FLAIR abnormality. Immediately following a SoC stereotactic biopsy procedure for confirmation of disease progression, BrainLab Flexible CED catheter(s) (7.5 mm tip) were placed, with at least one catheter proceeding along the same needle track as the SoC biopsy. Catheter placement was optimized to avoid ependymal surfaces by >0.5 cm as they offer no resistance to fluid flow and 2.0 cm from the closest sulcus, fissure, resection cavity, or cortical surface. Distal placement of the infusing tip within the tumor was also critical to avoid the pressure gradient from the tumor core to periphery to adjacent brain. Postoperative head CT was performed to evaluate for hematoma or pneumocephalus.

To allow for fibrin and clot deposition to occur, ¹⁸⁶RNL infusion was performed approximately 24 hours following catheter placement. ¹⁸⁶RNL was manufactured as previously described²⁰. Super-saturated potassium iodide (SSKI, 600 mg) was administered by mouth with water or juice prior to infusion. The infusion rate started at 1 µl/min and stepped to 20 µl/min (Table S6) using a syringe pump (Medfusion 3500 and 4000, Adepto Medical, Kansas City, MO). The total time of infusion was a function of flow rate, number of catheters used, and total volume infused (Table S6, Table S7, Table 1). During infusion, planar and tomographic images were collected using a dual-detector SPECT/CT camera. A sealed vial with known ¹⁸⁶RNL was positioned next to the patient at each time of image acquisition for radioactivity quantification. Dynamic images were acquired via real-time persistence scope for evaluation of focal accumulation of activity at the assumed tip of the catheter(s). When activity was observed to accumulate focally, that time was designated as the beginning of the planned therapeutic volume infusion to correct for dead space in the catheter line. Catheters deemed unsatisfactory (backflow along the catheter or spillage into adjacent CSF space) were stopped and the remaining volume switched to the remaining catheters.

To minimize backflow concerns³¹, we utilized the Brainlab catheter, which is both flexible and has a “step design” at the tip, avoided resection cavities and pial surfaces, and employed a slow ramping of infusion rates (Table S6). Furthermore, backflow was monitored during infusion using planar imaging as mentioned above, allowing changes to be made in real-time to ensure the planned administered dose was delivered in full.

The following reference was used as support:

- Wembacher-Schroeder, E. et al. Evaluation of a patient-specific algorithm for predicting distribution for convection-enhanced drug delivery into the brainstem of patients with diffuse intrinsic pontine glioma. *J Neurosurg Pediatr* **28**, 34–42 (2021).

- b. Report the AE by cohort using patient as unit of analysis to know what proportion of patients experienced those grades of AE in each cohort. The frequency of events using events are unit (Tables 4 and 5) can be supplemental.

We have revised the AE tables accordingly.

- c. Summary statistics on the absorbed doses per cohort.

We have provided a table for absorbed doses per patient/cohort (Table S8) and a figure of the summary per cohort (Figure 1).

- d. In Figure 1 (page 6), while the median Vd in cohort 6 is higher compared to all other cohorts, the maximum Vd is observed in a patient in cohort 5. Do the authors have an explanation for this?

We have revised the text in the Discussion section to state: *Following interim analysis of cohort 6, we observed a plateau from cohort 5 to cohort 6 in both dose distribution and absorbed dose. As a result of this finding, the recommendation of the Data Safety Monitoring Board (DSMB) was to expand cohort 6 to confirm tolerability of 22.3mCi administered dose in 8.8mL infusate volume as the Phase 2 recommended dose (RP2D) to target both bulk disease and adjacent microscopic disease in patients with one recurrence and tumor sizes of 20 cm³ or less. Once completed, and without the MTD being reached, all cohort data was analyzed with recommendation from the DSMB to proceed with Phase 2 utilizing this dose.*

- e. For survival analysis, was the dichotomization (<100 Gy or not) decided before the study onset, or is it a post-hoc analysis? Why was the survival analysis not done by assigned dose/cohort like ORR, but by absorbed dose?

We have tried to clarify the difference between the dose administered and radiation absorbed dose throughout the manuscript. While the study prospectively assigned patients to cohorts differing in administered dose, the study did not prospectively divide patients into radiation absorbed dose categories of > or <100Gy. It was not possible to know *a priori* what absorbed dose would be for a given administered dose as this is highly dependent on both distribution and retention. Indeed, this was determined on a case by case basis using post-infusion dosimetry measurements, as described in the Treatment section and then correlated with overall survival, among others. Survival analysis by Cohort – or administered dose – did not show any correlation with overall survival, as the enhancing tumor volume was required to be within the treatment field volume for a respective cohort, per the inclusion criteria. We therefore presented the survival data as we did, for all patients and the dichotomization, which we saw in our preclinical studies. We have revised the Results section to state: *Given finding from our preclinical studies identifying 100Gy as a threshold for survival benefit²⁰, patients were dichotomized by absorbed dose of 100 Gy.* Likewise, In the Introduction section, we state: *Furthermore, >100 Gy of ¹⁸⁶RNL was shown to eradicate grafted tumors and prolong overall survival, without clinical or microscopic evidence of toxicity²⁰.*

The following reference was used as support:

- Phillips, W. T. et al. Rhenium-186 liposomes as convection-enhanced nanoparticle brachytherapy for treatment of glioblastoma. *Neuro-Oncology* **14**, 416–425 (2012).

- f. For Table S4, did the authors examine the correlation between “Ratio of Treated to Total Tumor Volume”, “Total Dose in Distribution Volume”, and “Volume Administered” for the “Progression-Free Survival Accelerated Failure Time Model”? It is not clear why both of this need to be in the model.

We thank the reviewer for the question, but we are not completely sure of the intended meaning of the question. We believe this question to mean that some of the variables assessed are redundant. The first variable was to assess coverage independent of the dose administered, the second to assess the absorbed dose, and the third an ITT based on what was given rather than achieved.

- g. The trial accrued more males than females and mostly white patients; this could be discussed in terms of generalizability.

We thank the reviewer and note that the study population was reflective of the patient population in the centers participating. We have revised the text in a new Sex and Gender section to address the information regarding male/female accrual to state: *Sex and gender, as determined based on self-reporting, was not a factor in patient inclusion criteria, nor were there sufficient numbers of patients of either gender to make a meaningful sex- and gender-based conclusions in the study; as such, sex- and gender-based analyses have not been reported here.* Based on the small number of patients, we also extend this same rationale to race/ethnicity.